# Model systems to study tumor-microbiome interactions in early-onset colorectal cancer

Katharina M Richter[1,2], Marius Wrage[1,2], Carolin Krekeler [1,2], Tiago De Oliveira [3], Lena-Christin Conradi[3], Kerstin Menck[1,2,4] & Annalen Bleckmann [1,2,4 ✉]

## Abstract

**Colorectal cancer (CRC) is a major health problem, with an alarming increase of early-onset CRC (EO-CRC) cases among individuals under 50 years of age. This trend shows the urgent need for understanding the underlying mechanisms leading to EO-CRC development and progression. There is significant evidence that the gut microbiome acts as a key player in CRC by triggering molecular changes in the colon epithelium, leading to tumorigenesis. However, a comprehensive collection and comparison of methods to study such tumor-microbiome interactions in the context of EO-CRC is sparse. This review provides an overview of the available in vivo, ex vivo as well as in vitro approaches to model EO-CRC and assess the effect of gut microbes on tumor development and growth. By comparing the advantages and limitations of each model system, it highlights that, while no single model is perfect, each is suitable for studying specific aspects of microbiome-induced tumorigenesis. Taken together, multifaceted approaches can simulate the human body's complexity, aiding in the development of effective treatment and prevention strategies for EO-CRC.**

**Keywords** Cancer Cell Lines; Co-culture Systems; Early-onset Colorectal Cancer; Microbiome; Mouse Models
**Subject Categories** Cancer; Evolution & Ecology; Methods & Resources
Published online: 13 February

## Introduction

CRC is a major challenge for public health, ranking as the third most common cancer worldwide in 2024 (Pinheiro et al, 2024). CRC predominantly affects the elderly, with a current mean age at diagnosis of 74 and 77 years in both men and women, respectively (Murphy and Zaki, 2024). However, the number of affected patients under the age of 50 years has been constantly rising (Spaander et al, 2023), especially in high-income countries. Currently, EO-CRC accounts for up to 10% of all CRC cases and is expected to double during the upcoming 10 years in western countries. Due to the differences in disease characteristics and risk factors (summarized in Table 1), EO-CRC is considered as a distinct tumor entity compared to late-onset (LO)-CRC in individuals over the age of 50 years.

EO-CRC is more frequently observed in males (Ferrell et al, 2024) and among African American individuals compared to other ethnic or racial groups (Cercek et al, 2021). Up to 30% of EO-CRC cases show a familial or hereditary background with Lynch syndrome or familial adenomatous polyposis (FAP) as the most well-defined predispositions (Sameer, 2013). Hereditary mutations in the DNA mismatch repair system (MMR), or a hypermethylation of the *MLH1* promoter, lead to microsatellite instability (MSI) which is observed in up to 15% of CRC patients (Vilar and Gruber, 2010). MSI is more frequent in EO-CRC than LO-CRC and results in a different prognosis as well as treatment approach compared to microsatellite stable (MSS) tumors (Emile et al, 2024; REACCT Collaborative, 2022; Willauer et al, 2019).

The majority of EO-CRC cases, however, have no detected hereditary predisposition or MSI (Spaander et al, 2023). Mechanistically, the development of sporadic EO-CRC is explained by the traditional adenoma-carcinoma sequence of Vogelstein (Vogelstein et al, 1988). According to this model, the accumulation of mutations in the tumor suppressor genes *APC* and *TP53* as well as the oncogene *KRAS* induces the transformation of single colonic epithelial cells to aberrant crypts (Sameer, 2013). Over the years, the cells further degenerate into a benign pre-neoplastic lesion, which eventually transforms into malignant carcinoma due to the accumulation of additional genetic alterations. EO-CRC shows different molecular profiles compared to LO-CRC (Lu et al, 2023), including a distinct frequency of *BRAF*, *KRAS*, or *TP53* mutations. In general, a higher frequency of *TP53*, and lower frequency of *BRAF* mutations was reported in EO-CRC (Table 1), while contradictory observations have been reported for other genes, including *KRAS*.

Gene expression-based tumor subtyping is widely established and led to the introduction of four consensus molecular subtypes (CMS) in CRC (Guinney et al, 2015). In contrast to LO-CRC, CMS-1, which is associated with MSI and immune infiltration (Guinney et al, 2015), has been reported as the most common subtype in EO-CRC (Willauer et al, 2019). In line, EO-CRC tumors are often characterized by high tumor mutational burden (TMB) (Tang et al, 2024; Ferrell et al, 2024; Busico et al, 2024). Of note, a subgroup analysis observed that this difference was lost when excluding MSI tumors (Lieu et al, 2019; Lu et al, 2023). Likewise, the gene

[1]Department of Medicine A, University of Muenster, 48149 Muenster, Germany. [2]West German Cancer Center, University Hospital Muenster, 48149 Muenster, Germany. [3]Department of General, Visceral and Pediatric Surgery, University Medical Center Goettingen, 37075 Goettingen, Germany. [4]These authors contributed equally: Kerstin Menck, Annalen Bleckmann. ✉E-mail: annalen.bleckmann@ukmuenster.de

**Glossary**

**Adenoma**

Adenomas are benign (non-cancerous) tumors that originate from the glandular tissue lining certain internal organs, e.g., the gut. While adenomas themselves are not cancerous, they can evolve to cancer, if not removed.

**Carcinogen**

A carcinogen is any substance, organism, or agent that has the potential to cause cancer in living tissue. The mechanisms by which carcinogens induce cancer typically involve the alteration of cellular DNA, leading to mutations that disrupt normal cell growth and regulation.

**Carcinoma**

Carcinoma is a broad term that refers to any cancer that originates in epithelial cells, which line the surfaces of organs and structures throughout the body. Adenocarcinoma are a subtype of carcinoma originating specifically from glandular epithelial cells, which are responsible for the production and secretion of various substances, including hormones, enzymes, mucus, and other secretory products.

**Cell polarization**

Epithelial cell polarization refers to the asymmetry in cell shape, structure as well as distribution of molecular components in cells, creating an upper (apical), lateral as well as lower (basal) layer within a single cell. In the gut, the polarization of epithelial cells plays a crucial role in maintaining the integrity and function of the intestinal barrier (i.e., nutrient absorption, secretion, protection against pathogens).

**Colon**

The anatomy of the colon is divided into several distinct sections, each with specific functions and characteristics—right-sided and left-sided. The right-sided colon includes cecum and ascending colon, the left-sided colon includes transverse, descending and sigmoid colon.

**Consensus Molecular Subtypes (CMS)**

CMS is a classification system for colorectal cancer that categorizes tumors based on their molecular and genetic characteristics. It distinguishes between four distinct subtypes, each with unique biological features and clinical implications.

**Dysbiosis**

Dysbiosis refers to an imbalance in the composition, diversity, or function of the microbial community and has been associated with various diseases, including colorectal cancer (CRC). This imbalance can involve a reduction in beneficial microbes, an overgrowth of pathogenic species, or a loss of microbial diversity.

**Early-onset colorectal cancer (EO-CRC)**

EO-CRC is defined as CRC diagnosed in young individuals. Although there is no universal consensus on the age threshold, we categorize EO-CRC as occurring in individuals younger than 50 years. This classification distinguishes it from late-onset colorectal cancer (LO-CRC), which, according to our definition, occurs in adults >50 years of age.

**Germ-free and gnotobiotic mice**

Germ-free mice are completely devoid of any microorganism. Upon inoculation with a pure culture or pre-defined cocktail of microbes, gnotobiotic mice can be generated in which all microorganisms are well-defined and controlled.

**Gut microbiome**

The gut microbiome refers to the diverse community of microorganisms, including bacteria, archaea, fungi, and viruses, that inhabit the gastrointestinal tract.

**Microbial community**

A microbial community is an assemblage of diverse microorganisms that coexist and interact within a specific environment. These communities function as dynamic ecosystems, where microbes engage in complex interactions, such as competition, cooperation, and signaling, influencing both their environment and each other.

**Microfluidics**

Microfluidic devices typically consist of networks of microchannels and chambers designed to precisely control fluid flow and the positioning of cells within the system. This architecture enables the compartmentalization of different cell types while facilitating their communication through direct contact or via shared media, such as soluble signaling molecules.

**Microsatellite instability (MSI)**

MSI is a condition characterized by the accumulation of insertion or deletion errors in repetitive DNA sequences known as microsatellites. This phenomenon arises from a deficiency in the DNA mismatch repair (MMR) system, which is responsible for correcting errors that occur during DNA replication. When MMR is impaired, it leads to the creation of novel microsatellite fragments, resulting in a hypermutable phenotype.

**NOD *Rag2* γc mice (NRG)**

NOD *Rag2* γc mice (NRG) are highly immunodeficient. They harbor the IL2 receptor subunit gamma ($\gamma y$)$^{null}$ mutation and the *Rag2* knockout in the genetic background of the NOD mouse strain. These mice have no B and T cells and no functional NK cells. In contrast to NSG mice, NRG mice exhibit high tolerability to cytotoxic experimental therapies.

**NOD *scid* γ mice (NSG)**

NOD *scid* γ mice (NSG) are extremely immunodeficient as they carry two mutations on the NOD/ShiLtJ genetic background: The DNA repair complex protein Prkdc (*scid*) and the *IL2ry*$^{null}$ mutation. These mice have no B and T cells, no functional NK cells, no *IL2ry* expression and have an elevated sensitivity to DNA damage by cytotoxic agents.

**NOD/Shi-scid *IL2ry*$^{null}$ (NOG)**

The highly immunodeficient NOD/Shi-*scid-IL2ry*$^{null}$ (NOG) mice harbor the *scid* and *IL2ry*$^{null}$ mutations on the NOD/ShiJic genetic background. They have no B and T cells and no functional NK cells. Unlike in NGS mice, the *IL2ry* mutation in NOG mice produces *IL2ry* that is expressed and capable of binding cytokines but cannot transduce a signal.

**Obligate and facultative anaerobes**

Obligate and facultative anaerobes are classifications of microorganisms based on their oxygen requirements for growth and metabolism. Obligate anaerobes thrive only in anoxic environments due to their inability to tolerate oxygen, whereas facultative anaerobes exhibit metabolic flexibility that enables them to adapt to changing oxygen availability.

**Organoid**

An organoid is a three-dimensional (3D) miniaturized and simplified version of an organ that is produced in vitro from stem cells or stem-like cells, such as cancer cells. Organoids are designed to mimic the key functional, structural, and biological complexities of real organs.

**Patient-derived xenograft (PDX)**

PDX are mouse models where tumor tissues from patients are implanted into immunocompromised or humanized mice. This approach allows researchers to study the growth and behavior of human tumors in a living organism, maintaining the original tumors' histological and genetic characteristics.

**Transwell**

Transwells are small, cup-like components that consist of a membrane insert. Thereby, it creates two separate compartments, an upper and a lower one, within a culture dish.

mutation frequency was shown to depend on the racial or ethnic group, the MSI status as well as the TMB (Holowatyj et al, 2023; Lieu et al, 2019). Therefore, larger cohort studies considering these potential confounding factors are needed to deduce and confirm general trends for molecular tumor profiling in the different age groups.

In recent years, researchers have increasingly focused on lifestyle-related risk factors for CRC as potential explanations for the rise in EO-CRC. These factors comprise smoking, alcohol consumption, medication, obesity, physical inactivity, and dietary habits, including the consumption of highly processed foods, high-fat content and red meat. All of them are known to directly

**Table 1.** The key characteristics of EO-CRC compared to LO-CRC.

| Feature | Comparison EO-CRC vs. LO-CRC | Reference |
|---|---|---|
| Gender | Higher frequency of male patients in EO-CRC | Ferrell et al, 2024 |
| Race | Higher incidence of EO-CRC in Black/African American individuals | Cercek et al, 2021 |
| Obesity | Higher rate of obese and adipose patients in EO-CRC | Cercek et al, 2021; Liu et al, 2019 |
| Nutrition | Higher incidence of unfavorable nutrition among EO-CRC patients | Hur et al, 2021; Spaander et al, 2023 |
| Physical inactivity | Higher incidence of physical inactivity among EO-CRC patients | Spaander et al, 2023 |
| Inflammatory bowel disease (IBD) | Higher incidence of IBD-associated cancers in EO-CRC | Adnan et al, 2024 |
| Localization | Higher frequency of left-sided tumors (76–80% in EO-CRC vs. 61% in LO-CRC) | Cercek et al, 2021; Hashmi et al, 2023; Naing et al, 2024 |
| Histology | Poorer cell differentiation in EO-CRC | Cercek et al, 2021; Hashmi et al, 2023 |
| Germline predisposition / tumor syndromes | Higher prevalence of hereditary syndromes (up to 30% in EO-CRC vs. 10–15% in LO-CRC) | Mork et al, 2015 |
| CMS subtype | Higher frequency of CMS-1 subtype in EO-CRC | Willauer et al, 2019 |
| Microsatellite instability (MSI) | Higher frequency of MSI in EO-CRC | Willauer et al, 2019; Lawler et al, 2024; Emile et al, 2024; REACCT Collaborative, 2022; Ugai et al, 2023 |
| Tumor Mutational Burden (TMB) | Higher TMB in EO-CRC | Tang et al, 2024; Ferrell et al, 2024; Busico et al, 2024 |
| *BRAF* mutation | Lower frequency of *BRAF* mutations in EO-CRC | Ferrell et al, 2024; Lawler et al, 2024; Willauer et al, 2019; Holowatyj et al, 2023; Lieu et al, 2019; Ugai et al, 2023 |
| *TP53* mutation | Higher frequency of *TP53* mutations in EO-CRC | Ferrell et al, 2024; Holowatyj et al, 2023; Lawler et al, 2024; Lieu et al, 2019 |
| Tumor Immune System Interaction | Lower rate of age-dependent immune-senescence in EO-CRC | Yue et al, 2021 |
| Microbiome | Differences in composition and diversity between EO-CRC and LO-CRC | Kong et al, 2023; Weinberg et al, 2023; Zhou et al, 2024 |

influence the gut microbiome, which encompasses the entirety of archaea, fungi, and bacteria residing in the gut, ultimately leading to dysbiosis (Schulz et al, 2014; Murphy et al, 2019; Zhou et al, 2024). The gut microbiome plays an essential role in maintaining health. However, alterations in its composition can be detrimental and have been linked to various diseases, including CRC (Wong and Yu, 2023). All in all, CRC patients show a general change in their gut luminal microbiotic profile, including a lower α-diversity compared to healthy subjects (Kong et al, 2023; Ai et al, 2019; Barot et al, 2024). In contrast to LO-CRC, the gut microbiota species of young CRC patients differ significantly (Kong et al, 2023; Weinberg et al, 2023). While some bacterial species were found to be enriched in EO-CRC and are considered pro-oncogenic (e.g., *Bacteroides*, *Escherichia*, *Fusobacterium*, or *Porphyromonas*), others are believed to have a potentially protective effect (e.g., *Lactobacillus*, or *Roseburia*) (Abdullah et al, 2021; Gandini et al, 2024; Zhou et al, 2024). However, the definition of a consensus species signature for EO-CRC has been hampered by the plethora of lifestyle-related factors influencing the gut microbiome as well as methodological inconsistencies in the assessment of microbial profiles (She et al, 2024; Bars-Cortina et al, 2024).

Next to the gut lumen, microbiota have been detected inside the tumor tissue itself and can locally influence both carcinogenesis and anti-tumor immunity (Apostolou et al, 2011; Castellarin et al, 2012; Barot et al, 2024). For instance, the CRC-associated microbiome can modify the gene expression profiles of gut cells e.g., via epigenetic regulation or the induction of host immune responses, leading to the creation of a pro-inflammatory microenvironment (Jin et al, 2024). EO-CRC cases show a stronger microbial-host interaction at the tumor site compared to LO-CRC (Adnan et al, 2024). This is not surprising, as, on the one hand, the function of the immune system deteriorates with age, a process called immunosenescence (Yue et al, 2021). This involves the exhaustion of progenitor cells, reduced production of adaptive immune cells and altered secretion of matrix metalloproteinases and pro-inflammatory cytokines (e.g., IL1β and TNFα), ultimately resulting in a low-level inflammation and a dysfunctional immune response to pathogens (Doles et al, 2012; Czesnikiewicz-Guzik et al, 2008; Deng et al, 2004). On the other hand, aging not only affects the host but is also observed in the gut microbiome itself where it causes age-related dysbiosis with implications for the host immune system (comprehensively reviewed in (Bosco and Noti, 2021)). The comparison of the immune landscapes in EO-CRC and LO-CRC is a matter of current research (Du et al, 2023; Griffith et al, 2023; Andric et al, 2023) and will hopefully improve our understanding of the mutual effects of host-microbiome interactions during aging and their functional relevance for EO-CRC development.

Despite rising evidence that differences in microbiome composition and function are crucially involved in cancer initiation and progression, the precise mechanisms by which certain microbes contribute to EO-CRC remain largely unclear—often due to the lack of suitable model systems. In this review, we provide a comprehensive overview of the available approaches and how they could be used to address this problem. Potential advantages, disadvantages and applications of the single approaches are

Table 2. Model systems and their advantages and disadvantages to study the influence of the microbiome in EO-CRC.

| Model system | Benefits | Limitations | Duration of microbiome co-cultures |
|---|---|---|---|
| Carcinogen-induced mouse models | - Model for colitis-induced EO-CRC available<br>- Comparably easy, fast, and cost-effective set-up<br>- High reproducibility<br>- Useful for studying the effect of distinct dietary components<br>- Useful for studying early- or late-stage tumorigenesis | - Moral and ethical issues<br>- High latency prevents studying tumor development in young animals<br>- Not suitable to study tumor invasion and metastasis<br>- Carcinogens influence the microbiome and the whole gastrointestinal system | Long-term |
| Genetically engineered mouse models (GMMs) | - Models for FAP and MSI available<br>- Useful for studying the specific contribution of selected genes<br>- Useful for studying tumor initiation, progression and metastasis | - Moral and ethical issues<br>- High latency in some models prevents studying tumor development in young animals<br>- Different tumor location in some models compared to human patients<br>- Expensive and time-consuming to develop and maintain genetic lines | Long-term |
| Tumor transplantation models | - PDX maintain the genetic characteristics of the individual patient and allow studying patient-specific effects on the microbiome<br>- The orthotopic injection of GMM tumor cells allows studying tumor-microbiome interactions in young, immune-competent animals<br>- Useful for studying tumor growth and progression | - Moral and ethical issues<br>- Lack of murine EO-CRC cell lines<br>- Lack of immune system and hampered tumor–stroma interactions in xenograft models<br>- Lack of local microbiome in s.c. transplantation models | Mid-term<br>Long-term |
| Germ-free and gnotobiotic models | - Comparably easy and fast set-up for antibiotics treatment<br>- Humanized models available<br>- Useful for studying the effect of single microbes as well as complex microbial communities<br>- Can be combined with FMT from EO-CRC patients | - Moral and ethical issues<br>- Expensive and time-consuming to develop and maintain germ-free mice<br>- Germ-free mice harbor defects in the gut and immune system<br>- Antibiotics cannot eradicate the whole microbiome<br>- Antibiotics cause several side effects<br>- Require combination with other CRC models to induce tumor growth | Long-term |
| Cell lines | - Low-cost and high-throughput model<br>- Controllable conditions<br>- Easily available for genetic engineering<br>- Versatile for use in distinct in vitro and in vivo settings<br>- Useful for mechanistic studies on single-cell level | - Lack of cell heterogeneity<br>- Lack of cell polarization, mucus production, and oxygen gradients under normal 2D conditions<br>- Limited availability of cell lines from young individuals<br>- Artificial and immortalized culture conditions with limited translatability to human patients | Short-term |
| Transwell-based models | - Low-cost and high-throughput model<br>- Controllable conditions<br>- Support cell polarization<br>- Create an oxygen low apical and oxygen high basolateral compartment<br>- Useful for studying bacterial adhesion and invasion (up to 24 h) | - Lack of cell heterogeneity<br>- Lack of oxygen gradients<br>- Lack of mucus production and thus missing tissue barrier<br>- Static<br>- Rapid bacterial outgrowth | Short-term |
| Microfluidics | - Controllable conditions<br>- Support cell differentiation, mucus production, and crypt formation<br>- Possibility of concentration gradients and addition of multiple cell populations<br>- Useful for studying direct or indirect host-bacteria interactions in longitudinal co-cultures (up to 1 week) | - Lack of cell heterogeneity<br>- Lack of standardization<br>- Technically challenging<br>- Require custom-made components<br>- Require long establishment period | Short-term<br>Mid-term |
| Organoids | - Replicate the natural gut tissue architecture in a 3D environment<br>- Consist of heterogeneous cell populations<br>- Potential for automation and standardization<br>- PDOs maintain the genetic characteristics of the individual patient<br>- Useful for personalized drug screens and mid-term co-cultures (up to 72 h) | - Technically challenging<br>- Time-consuming<br>- Expensive growth factors<br>- Require microinjections for studying bacteria-host interactions at the apical site<br>- Lack of microenvironmental cells in most models | Short-term<br>Mid-term |

**Table 2.** (continued)

| Model system | Benefits | Limitations | Duration of microbiome co-cultures |
|---|---|---|---|
| Ex vivo models | - Maintain the spatial gut architecture<br>- Consist of heterogeneous cell populations, including microenvironmental cells<br>- Presence of mucus layer<br>- Can be obtained from different gut compartments<br>- Tissues maintain the genetic characteristics of the individual patient<br>- Useful for studying barrier function in real tissue (up to several h)<br>- Useful for studying gut segmental contractions (up to 48 h)<br>- Useful for studying spatiotemporally controlled tumorigenesis at the single-cell level (up to several weeks) | - Ethics and regulatory issues<br>- Technically challenging<br>- Require experimental optimization for co-culture applications<br>- Lack of oxygen and nutrient supply across the tissue<br>- Lack of automation and standardization | Short-term<br>Mid-term<br>Long-term |

Co-culture and/or study duration: Short-term ≙ < 24 h; Mid-term ≙ < 1 week; Long-term ≙ > 1 week.

highlighted in Table 2. We thereby aim at helping researchers in selecting the most appropriate model for their specific question and enable them to shed further light on the complex interactions between the gut microbiome and colorectal cells in the context of EO-CRC.

## In vivo assays

Preclinical animal models play an important role in understanding the influence of the microbiome in the development and progression of CRC. The ideal model should reflect the whole process of cancer development, starting with precancerous adenomas to an invasive metastasizing cancer, while allowing researchers to evaluate the microbial communities and immune populations present in the animal's gut. However, due to the complex and not fully understood interactions between genetic, microbial and environmental factors resulting in EO-CRC development, the matching of all required aspects is hard to be obtained. Researchers who want to study the role of the microbiome in vivo need to carefully choose their model system based on factors such as latency, intestinal location and molecular signature, as not all of them accurately match human EO-CRC.

The following section will mainly focus on mouse models. While zebrafish and drosophila models exist as well and provide unique ways to study CRC development using imaging approaches, their use for microbiome studies is limited (Wang et al, 2022; Adams et al, 2021). The gastrointestinal system of rats is more similar to that of humans compared to mice, however, the established rat CRC models are few and show a latency in tumor development of several months (Miyamoto and Tani, 1989; Femia et al, 2015a, 2015b). Mice, on the other hand, offer several advantages for studying CRC, such as precise genetic modifications, modest induction of tumor growth, rich microbiome and sufficient anatomical and physiological similarities to humans. In addition, mouse experimental setups frequently need to be performed at high hygienic and controlled conditions, as the use of both germ-free and gnotobiotic mice is essential in understanding how the presence - or absence - of certain microbes can directly influence CRC (Jans and Vereecke, 2024). This review will align with the scientific consensus on age classification of mice, which proposes the equivalence of mouse lifespan to human maturation as follows: (1) young from 3 to 6 months; (2) middle-aged from 10 to 14 months; and (3) old at 18 months and beyond (Dutta and Sengupta, 2016; Flurkey et al, 2007).

## Carcinogen-induced CRC models

Carcinogen-induced models have been used to study the progression of early lesions, such as aberrant crypt foci, to adenomas and carcinomas, as well as the influence of dietary supplements, meat consumption and microbiota on colorectal carcinogenesis (Neto et al, 2023). They are based on the application of enterotoxic, mutagenic and alkylating agents, such as azoxymethane (AOM), heterocyclic amines (HAA) or N-Methyl-N'-nitro-N-nitrosoguanidine (MNNG) (Bürtin et al, 2020; Subapriya et al, 2005). Colorectal tumors induced by HAA, which naturally emerge during the cooking of fish and meat at high temperatures, display a faster growth upon supplementation of mice with a high-fat diet, underlining the importance of nutrition for tumorigenesis in the gut (Chen et al, 2017).

The major advantages of carcinogen-induced models are their high reproducibility, the comparatively low costs and the combinability with dietary components. Thus, these models enable researchers to mirror the lifestyle circumstances present in their specific population of interest. Moreover, depending on the dosage and timing of carcinogen exposure, researchers can study early- and late-stage carcinogenesis (Neufert et al, 2021; Subapriya et al, 2005). The disadvantages of carcinogen-induced models are that differences in strain origin, gender and housing conditions can significantly interfere with tumorigenesis, eventually providing contrasting results (De Oliveira et al, 2019; Elrod et al, 2005). Furthermore, the induced tumors rarely show invasive behavior and metastatic spread, but rather form synchronous adenomas, indicating that this model is rather suitable for studying tumorigenesis and not tumor progression to metastatic CRC (Rosenberg et al, 2009). Another challenge is the high latency of 24–50 weeks for the development of tumors (Bürtin et al, 2020), eventually limiting the use of carcinogen application to study EO-CRC in young animals. Combining the carcinogens with a colitis-inducing reagent such as dextran sodium sulfate (DSS) can fasten

tumor growth, however the treated mice reproducibly display signs of colitis (De Robertis et al, 2011). Since colitis-induced CRC shows distinct clinical features and disease pathogenesis compared to sporadic CRC in human patients, DSS treatment cannot be recommended to study EO-CRC, unless the focus is to investigate how pro-inflammatory conditions (e.g., IBD) affect EO-CRC development. Furthermore, different studies have described an influence of carcinogens on the microbiome of the animals and vice versa (reviewed in (Ambalam et al, 2016)). Overall, carcinogen-induced CRC models offer several advantages that make them valuable tools for understanding general CRC mechanisms; however, they do not adequately reflect EO-CRC due to the high latency in tumor development, the lack of invasive behavior, and the effect of the carcinogens on the gut microbiome.

## Genetically engineered mouse models (GMMs)

GMMs have played a central role in the identification and understanding of CRC initiation and development. Different GMMs have been established to recreate sporadic tumor development as observed in EO-CRC. The most commonly introduced mutations, $Apc^{Min/+}$ and $Apc^{\Delta 716}$, induce the formation of multiple intestinal neoplasms (Min) within several weeks after birth, thus modeling the first steps of cancer development in young mice. A major drawback is that the adenomas predominantly arise in the small intestine, and rarely in the colon, where they primarily occur in human EO-CRC patients.

The deletion or mutation of $Apc$ leads to the nuclear accumulation of β-catenin and constitutive activation of the Wnt pathway, resulting in dysregulated cell proliferation (Ren et al, 2019). Inactivation of $APC$ is accountable for FAP, and more than 80% of sporadic CRC patients harbor an aberrant $APC$ gene (Dhir et al, 2008). Based on the multistep model of CRC carcinogenesis, the initial adenomas in $Apc^{Min/+}$ mice can transform into malignant tumors upon loss of the remaining $Apc$ allele and accumulation of additional tumor-driving mutations. However, this mostly occurs at a tardy time point resulting in a high latency until tumor development and limiting the potential of studying later stages of carcinogenesis in young mice (Metzger et al, 2019). Li et al showed that $Apc^{Min/+}$ mice gavaged with gut microbiota from CRC patients show higher intestinal barrier damage and faster progression of adenomas than their littermates fed with feces from healthy controls, suggesting a microbiome-driven influence on CRC carcinogenesis (Li et al, 2019).

Combining $Apc$ mutations with deletions of the TGF-β signaling mediators $Smad3$, $Smad4$ or $Smad3/4$ in double, triple or cell-specific LoxP/Cre knockout mice further accelerates tumor formation but the majority of tumors continue to arise predominantly in the small intestine (Neto et al, 2023; Bürtin et al, 2020). Both issues were addressed in $Apc^{fl/fl}$ $Smad4^{fl/fl}$ $Car1^{CreER/+}$ $Kras^{LSL-G12D/+}$ $Tp53^{KO}$ mutant mice. Here, the mutations lead to aggressive carcinoma formation in the cecum and the proximal colon with lymph node invasion within 1 month, ultimately causing a high mortality (Tetteh et al, 2016). Metastatic spread can also be observed in GMMs harboring $Braf^{LSL-V637E/+}$ or $Tp53^{R172H/+}$ or $Kras^{G12V/+}$ and $Pten^{fl/fl}$ deletions in around 20–40% of the animals, however, they usually do not occur in young animals (Rad et al, 2013; Davies et al, 2014; Reischmann et al, 2020). Those genetic alterations undoubtedly contribute to the etiology of CRC,

triggering direct changes not only in the intestinal epithelium but also in its crosstalk with the immune system and the gut microbiota, pre-dispositioning tumorigenesis (Yuan et al, 2022). Nevertheless, single or double mutants cannot fully mimic the EO-CRC phenotype due to a high latency and different tumor location compared to human patients. Breeding, maintaining and testing of $Apc$ mice and triple or quadruple mutants is difficult, expensive, time-consuming and the influence of the pre-existing mutations on the microbiota has not been well understood yet.

In order to homozygously delete specific CRC driver genes at a defined time point to induce local tumor growth, the tamoxifen-inducible Cre-loxP system is an option. The intraperitoneal or local injection of a tamoxifen/corn oil solution into mouse lines harboring respectively floxed alleles, leads to a spatiotemporally controlled activation of the Cre-ERT2 recombinase and subsequent gene inactivation (Li et al, 2022). This system also allows targeting genes associated with pre-adult lethality (Valny et al, 2016) and showed no difference in effectiveness in young or old mice (Kellogg et al, 2023). Using the Villin-Cre expression system, recombination can be specifically directed to intestinal epithelial cells (Pinto et al, 1999; Madison et al, 2002). In 2019, Radhashree et al published their results for a mouse strain with tamoxifen-inducible epithelial gut-specific $CDX2P$-Cre recombinase combined with $Apc^{fl/fl}$ and $Kras^{+/fl}$. These mice developed a highly malignant tumor, which resembled MSS characteristics within 4 weeks (Maitra et al, 2019). To study the relationship between MSI and microbiome changes in EO-CRC, the use of Villin$^{CreER}$ Mlh1$^{fl/fl}$ mice (Leach et al, 2021) could be a promising model, although it may be necessary to combine the strain with other tumor mutation-carrying GMMs or with a carcinogen-induced model to reduce the latency for tumor development. A confounding factor of the Cre-loxP system is that even a single, local injection of tamoxifen (4-OHT), can cause a possible systemic effect in some strains, leading to the formation of tumors also in other parts of the intestine (Hou et al, 2021; Ramesh et al, 2021). Therefore, the pattern of Cre activity and the percentage of Cre-expressing cells should be tested prior to the experiment using different doses of a single local injection.

Another interesting, but technically challenging, approach is the local induction of tumor growth by surgically infecting the colon of conditionally floxed mice with Cre recombinase-carrying adenoviruses, which target the basal colonic crypt cells and lead to the homozygous deletion of floxed CRC driver genes (Hung et al, 2010). Tumor growth can thus be initialized at a well-defined time point and region of the colon of conditional knockout mice, e.g., $Apc^{fl/fl}$, $Kras^{fl/fl}$ or $Tp53^{fl/fl}$. Visible adenoma formation can be observed after 2–4 weeks and progression to invasive adenocarcinoma is set after 12–16 weeks, thus occurring in middle-aged animals rather than in young ones. Alongside reduced animal survival (Betzler et al, 2017), the influence of tamoxifen toxicity or the adenovirus-mediated Cre knockout may trigger a host immune response which could significantly alter the gut microbiome (Prost, 2001). This effect therefore needs to be carefully evaluated for EO-CRC microbiome studies.

The combination of GMMs and carcinogen treatment could be another interesting approach to analyze the role of the microbiome in EO-CRC. In 2019, Souris et al published their results for a mouse strain with epithelial gut-specific $CDX2P$-Cre and heterozygously floxed $Apc$ in combination with AOM treatment. Their experiments showed promising results for short-term EO-CRC studies, without technically demanding and time-consuming protocols (Souris et al, 2019).

Altogether, good GMM models have already been developed for investigating the impact of microbiome dysbiosis on CRC development. Researchers should carefully choose GMM models as some have a pre-existing age effect or tumor formation outside of the colon. Although only few studies have directly applied them to study the molecular and microbial processes involved in the EO-CRC setting, the Cre recombinase-mediated induction of colon-specific mutations via tamoxifen is an excellent approach to model EO-CRC in mice. It results in rapid and aggressive tumor growth and can be combined with dietary intake or fecal microbiota transplantation (FMT) studies to investigate the impact of specific genes and/or microbiota on EO-CRC development and growth.

## Tumor transplantation models

Transplantation models profit from a relatively easy and fast set-up. The heterotopic subcutaneous (s.c.) transplantation of murine CRC cells (e.g., CT26, MC38) into recipient mice is the most commonly used syngeneic mouse model, offering the opportunity to study tumor growth in animals with a functioning immune system (Greenlee and King, 2022; Neto et al, 2023). It has been shown that the injection of MC38 cells into young C57BL/6 mice (3–4 months) resulted in faster and more aggressive tumor growth compared to middle-aged animals (12–14 months). The different growth rates were not observed in immune-deficient young versus old RAG1[null] mice, underlining the central role of the immune system for CRC tumorigenesis (Ershler et al, 1984). However, as the injected tumor cells rarely metastasize and a direct interaction between the gut microbiota and tumor cells at the flank is not observed, this model would not be advised for evaluating the impact of the microbiome on EO-CRC growth. Instead, orthotopic injections are more suitable to study the impact of specific microbial communities on CRC in vivo. Murine CRC cell lines or cells harvested from GMMs used for transplantation into immune-competent young mice could be easily genetically modified to understand the contribution of specific genes or pathways (Evans et al, 2019). However, currently, no murine cell lines accurately recapitulating EO-CRC are available. Moreover, although such an approach would allow studying the growth and progression of tumor cells in their natural environment, including their interaction with the microbiome, researchers have refrained from using them due to the difficult transplantation procedure and low engraftment rate. Newly generated methods such as the endoscopy-guided minimally invasive model developed by Chen et al could help to overcome these challenges (Chen et al, 2020).

Another tumor transplantation approach is the use of xenografts. Methodically, the implantation of human tumor cells into immune-compromised mice (e.g., nude or severe combined immune-deficiency mice, such as NOD *scid* γ mice (NSG) or NOD *Rag2 γc* mice (NRG)) allows studying tumor growth in a physiological 3D environment. Depending on the implantation site and the number of tumor cells used, the mice show fast tumor growth rates with occasional metastasis (Betzler et al, 2017). The preferred age of host mice for engraftment is 5–6 weeks as these mice provide a stable and conducive environment for tumor formation and growth (Quintana et al, 2008). In addition, also in our experience with CRC xenografts, implantation into young NRG and NOD *Scid* animals shows better engraftment than in old ones. We hypothesize that this is again due to better microenvironment

implantation conditions, such as good perfusion and faster cellular turnover during the regeneration process after implantation (Edelmann et al, 2024).

A more sophisticated approach with increasing popularity is the patient-derived xenograft (PDX) model, which is thought to recapitulate the genetic and molecular heterogeneity of human cancers and has been increasingly used in personalized medicine studies to test novel therapeutic strategies (Bürtin et al, 2020). Injecting patient-derived CRC stem-like cells into the submucosa of the ascending colon of immunocompromised mice, Trivieri et al showed that the *BRAF* status of CRC cells has a significant impact on the gut microbiome profile (Trivieri et al, 2020). These PDX mice can further be used to study metastasis, as spreading cells show histopathological similarities to CRC patients (Li et al, 2022). Hence, PDX models can represent valuable tools to examine the influence of patient-specific CRC subclones on local microbial communities. Nevertheless, important limitations have to be considered: First, the use of PDX models is hampered by a longer establishment period compared to the direct injection of established cancer cell lines, and although the co-injection of endothelial colony-forming cells along with tumor cells could serve as a rational approach to accelerate tumor growth in this model (Kwon et al, 2021), this still is a relevant restriction. A second, and even more important drawback is that PDXs must be established in immune-deficient mice, and thus lack the complex and valuable interaction between the distinct immune cell populations, the tumor cells, and the microbiota. In addition, many of these models, such as the severely immunodeficient NSG and NRG mouse strains, exhibit diminished cytokine release, which can further disrupt signaling crosstalk between immune cells and intestinal epithelial cells (Kitsera et al, 2023). This alteration in immune signaling may lead to unpredictable modifications in their interactions with the microbiome. Lastly, the interaction between human tumor cells and the newly formed murine tumor stroma must be interpreted with caution due to the species divergence (Morgan, 2012). By implantation of tumor organoids from Villin-Cre x LSL*Kras*[G12D] mice into the rectal or distal colon of gender-matched recipient mice, some of these issues have been circumvented. In this regard, Felche et al have developed a heterogeneous single tumor capable of metastasis formation, which will enable the analysis of the direct interactions between the tumor microenvironment and the microbiome in immunocompetent young mice in the future (Felchle et al, 2024). In summary, upon orthotopic injection into young immunocompromised mice, PDX models allow studying the patient-specific interplay of EO-CRC cells with the local micro-biome for tumor growth and spreading, however, at the limitation of a missing immune system and the species difference hampering tumor–stroma interactions. The orthotopic injection of GMM-derived tumor cells into immunocompetent young mice could generally circumvent these problems at the cost of a long and sophisticated establishment period and the challenge of translating the findings to human patients.

## Germ-free and gnotobiotic models

In order to specifically study and modify the microbiome in vivo, germ-free mice remain the best-controlled model, in particular for long-term microbial transplantation studies (Jans and Vereecke, 2024). These mice are bred in specialized facilities and maintained in sterile

isolators, making them completely devoid of all microorganisms, and thus allow studying the effect of an absent microbiome on the biological process of choice. Upon colonization with one specific microbe, minimal microbial consortia, or a whole functional microbiome, e.g., via fecal microbiota transplantation (FMT) of human stool samples, (humanized) gnotobiotic mouse models with a defined microbiome can be created. In this regard, the timing of FMT is an important factor for optimal colonization (Hansen et al, 2012). Studies have shown that around 60–85% of bacteria were concordant at the genus level when comparing human donor and murine recipient stool samples 2 weeks after FMT, while the relative abundances of the individual microbes remained largely similar (Shaikh et al, 2022; Turnbaugh et al, 2009). Recent advances have allowed to reconstitute immune-deficient NOG mice with human hematopoietic stem cells prior to FMT of human stool samples to create a dual-humanized model in young mice (Ka et al, 2023).

FMT, when combined with suitable EO-CRC models, can yield valuable insights into the impact of whole microbiomes and single bacterial strains on tumor development and growth. Of note, FMT from old (18 months), but not young (11–14 weeks), conventional mice into young (12–14 weeks) germ-free mice results in a low grade inflammation (Fransen et al, 2017) and can thus predisposition the animals for carcinogenesis. Indeed, Crossland et al performed FMT from young (6 weeks) and old (72 weeks) mice into young (8 weeks) recipient mice. After initiation of tumorigenesis with AOM, the authors showed that FMT from older mice led to higher colonic cell proliferation, inflammation and tumor abundance, indicating age-dependent differences in the gut microbiome and its impact on tumorigenesis (Crossland et al, 2023). Likewise, Schulz et al have demonstrated young mice carrying an intestinal-specific $Kras^{G12D}$ mutation and having been fed with a normal diet to exhibit increased tumorigenesis after FMT from high-fat diet-fed mice, indicating that the microbiome can transmit high-fat diet-induced tumor progression in genetically predisposed animals (Schulz et al, 2014).

Despite the huge potential of germ-free models, only few are commercially available, as they must be kept in specialized facilities, require a constant monitoring and are thus very time-consuming, difficult and expensive to establish and maintain. In addition, germ-free mice display anatomical and cellular abnormalities in the gut and have an underdeveloped immune system (Round and Mazmanian, 2009), confounding factors which should be considered when applying the model to study EO-CRC. An alternative is to modify the microbiome by treating the mice with a combination of broad-spectrum antibiotics (Reikvam et al, 2011). While this approach allows studying microbiome changes in animals with prior microbial stimulation and priming of the immune system in early life (Kennedy et al, 2018), the use of antibiotics has several disadvantages, including a broad-scale effect on the gut itself, other organs and their microbiomes (Jans and Vereecke, 2024; Oh et al, 2016; Dessein et al, 2020) as well as delayed gastrointestinal and colonic motility and metabolism (Ge et al, 2017). A limiting factor is that antibiotics do not eradicate all bacterial species, specifically select for resistant bacteria (Lundberg et al, 2016), and do not target the other components of the gut microbiome (i.e., viruses, fungi), thereby favouring fungal outgrowth (Kim et al, 2014). New tools for microbiome editing are currently being developed which allow targeting and modifying selected bacterial strains within microbial communities of the gut (Neil et al, 2021; Rubin et al, 2022; Brödel et al, 2024) and could circumvent some of these constraints in the future.

Taken together, germ-free and gnotobiotic models as well as microbiome editing approaches are valuable tools to study the impact of specific microbes and whole microbiome communities on EO-CRC development and progression in young hosts in vivo. However, they rely on the combination with any of the established CRC models to predispose and induce EO-CRC formation.

## In vitro and ex vivo assays

Although the significance of the microbiome for CRC development has been acknowledged, the molecular effects of specific bacterial strains that might drive the transformation of colorectal cells is still scarce. In vitro and ex vivo models serve as an essential tool to address this gap and study tumor-microbiome interactions at the single-cell level in a controlled environment. A schematic representation of the most commonly used co-culture models, which will be discussed in this chapter, is provided in Fig. 1.

## Cell lines

Cell lines serve as a good model system to study causal and mechanistic relationships up to single-cell level and are easy to obtain, culture and modify. A multitude of commercially available and molecularly well-characterized human and murine colorectal (cancer) cell lines has been established and serves as the basis for most in vitro and/or mouse tumor transplantation studies (Berg et al, 2017; Ahmed et al, 2013; Ho et al, 2021). However, as most of the frequently used human CRC cell lines (e.g., Caco-2, SW48, SW620) originate from elderly individuals, they are not the optimal choice for studying EO-CRC. While a limited number of cell lines from young individuals is available (Table 3), it is important to note that the molecular characteristics of EO-CRC, such as microsatellite and mutation status, must also be considered when selecting appropriate models.

To investigate the influence of the microbiome on cellular transformation, researchers can access various normal colorectal epithelial cell lines (Table 3). Of note, some of these cell lines are of embryonic origin and might not possess the same characteristics as adult colorectal cells (Brumbaugh et al, 2023). Alternatively, primary adult colon epithelial cells (HCoEpiC) can be obtained from different providers, however, they are sourced from different donors, and the age and gender of the donors can vary. Unfortunately, the number of studies employing these cell lines is scarce and therefore they remain poorly characterized.

Taken together, cell lines are a valuable tool for genetic engineering studies to investigate the impact of specific genes or microbiota on colorectal (tumor) cell function. However, there is a significant need for the establishment and comprehensive characterization of additional cell lines from younger individuals, including the comparison with patient-derived primary cells. As most cell lines have been immortalized and have adapted to the artificial 2D culture conditions, they might not adequately represent the physiological properties of cells in vivo, thus complementary studies using in or ex vivo models are generally required for translating cell line-based observations to human patients.

## Setting up colon–microbiome co-cultures: general considerations

Co-culturing human colorectal cells with gut bacteria poses several challenges due to the distinct environmental requirements. On the

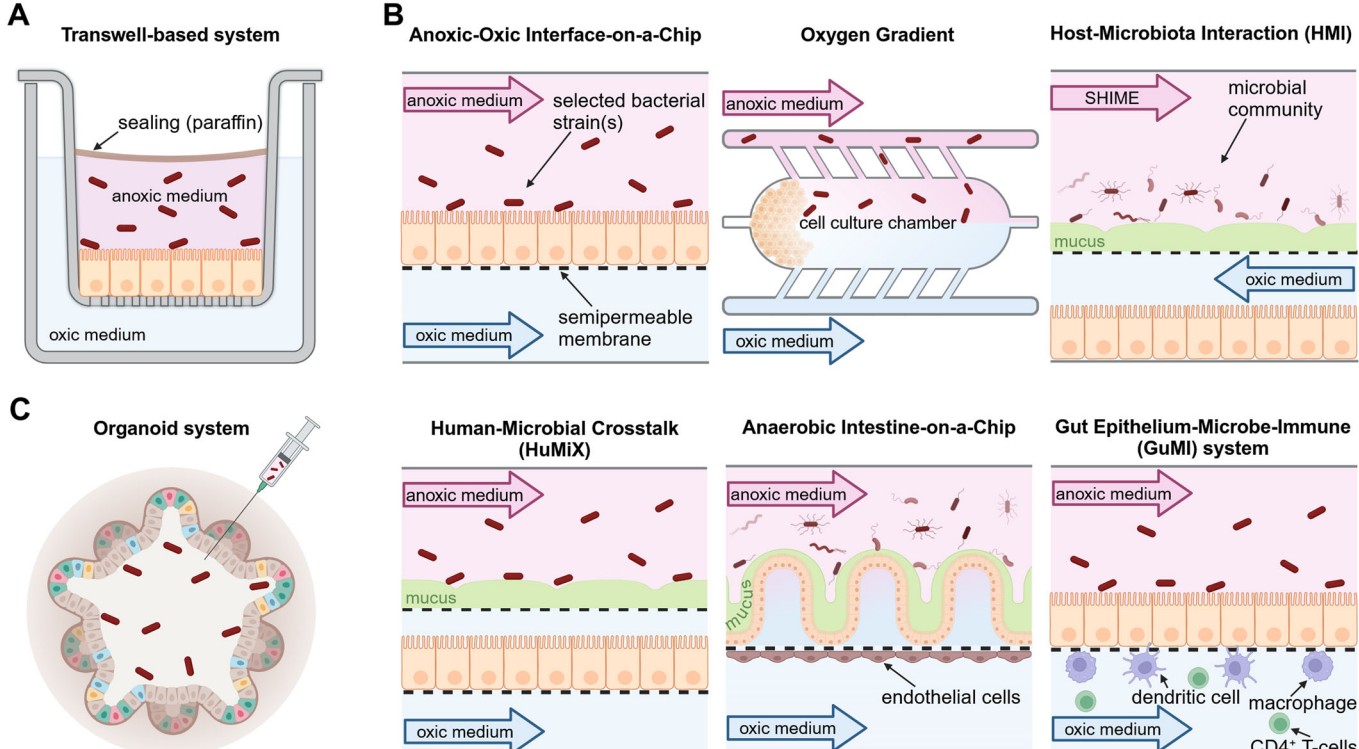

**Figure 1. In vitro co-culture systems.**

Schematic representation of a selection of frequently used in vitro co-culture systems to study colon–microbiome interactions. (A) Transwell-based systems are static systems, creating apical-anaerobic and basolateral-aerobic compartments, enabling the study of bacteria (red rods) adhesion, invasion, and interactions with host cells (orange). (B) Microfluidic devices provide constant medium flow, allowing for extended co-culture durations, while accommodating different oxygen needs for host cells and bacteria. The arrows represent the direction of flow. (C) Organoid cultures enable the study of interactions between gut microbes (red rods) and various intestinal cell types in a physiologically relevant 3D system. Often, bacteria are microinjected into the lumen of the organoids to enable their direct contact with the apical side of the colon epithelial cells.

one hand, eukaryotic host cells and bacteria are cultivated in different growth media, on the other hand both differ in their requirement for oxygen. While an adaptation of eukaryotic cells to the rather minimal bacterial growth media is often impossible, specific nutrients and components in the human cell culture media can affect bacterial growth rates and metabolic activity (Maier et al, 2017). Co-culture systems with separate chambers for the respective cells, or media, might be one possibility to address this challenge. While human colorectal cells frequently require oxygen, gut bacteria are typically obligate anaerobes (Ulluwishewa et al, 2015). To overcome this problem, some studies have been designed in short-term settings only (up to 4 h) (Robson et al, 2023). Although this might be sufficient to model the adhesion of bacteria to the cells, the capabilities to monitor cellular transformation upon prolonged contact with the microbiota are jeopardized. An alternative approach is the use of dead bacteria or bacteria-depleted conditioned medium to stimulate host cells (Maier et al, 2018; Hwan Choi et al, 2008), which is useful to study one-way intercellular communication or evaluate the contribution of bacterial components, secreted factors or extracellular vesicles. However, it disregards the dynamic secretion of factors by living bacteria upon contact with the host. Indeed, different effects on pro-inflammatory Toll-like receptor (TLR) signaling in HEK293 cells co-cultured with viable or dead bacteria were observed (Maier et al, 2018).

Polarization of colorectal epithelial cells is pivotal in the gut and essential for maintaining the integrity of the intestinal epithelial barrier and regulating microbial interactions at the apical side (Ali et al, 2020). Many receptors involved in microbial sensing, such as TLRs, show a differential expression pattern on the apical and basolateral side of polarized epithelial cells. This distribution affects how cells respond to microbial stimuli (Hug et al, 2018). It has been shown that in vitro polarized cells are able to develop in vivo-like microvilli structures on the apical side and secrete mucus, which influences microbial interaction and colonization (Jalili-Firoozi-nezhad et al, 2019). A thick mucus layer, like in the intestinal lumen, can prevent bacteria from directly interacting with host cells (Cornick et al, 2015). Therefore, cell polarization is an important requirement when designing a representative model system to study tumor-microbiome interactions.

## Transwell-based co-culture models

Transwell inserts consist of a permeable membrane support suspended in a well of a tissue culture plate, creating two compartments, an apical and a basolateral one (Maier et al, 2018). Such a set-up allows the cells to access nutrients from both sides and promotes the formation of epithelial apical–basal polarity

**Table 3. Colorectal (cancer) cell lines originating from young human individuals.**

| Cell line | Age, sex, race | Source | MS-status | Mutations | Reference |
|---|---|---|---|---|---|
| HT-29 | 44, f, Caucasian | Adenocarcinoma, colon, Dukes' C, primary tumor, CMS3, colon-like | MSS | *APC* E853* T1556fs*3<br>*TP53* R273H<br>*KRAS* wt<br>*BRAF* V600E<br>*PIK3CA* P449T<br>*SMAD4* Q311* | Fogh and Trempe, 1975; Berg et al, 2017 |
| HCT116 | 48, m, Caucasian | Carcinoma, ascending colon, Dukes' D, primary tumor, CMS4, undifferentiated | MSI | *APC* wt<br>*TP53* wt<br>*KRAS* G13D<br>*PIK3CA* H1047R<br>*CTNNB1* S45del<br>*CDKN2A* R24fs*20, E74fs*15<br>*BRCA2* I2675fs*6 | Ahmed et al, 2013; Berg et al, 2017 |
| SW480 | 50, m, Caucasian | Adenocarcinoma, descending colon, Dukes' B, primary tumor, CMS4, undifferentiated | MSS | *APC* Q1338*<br>*TP53* R273H, P309S<br>*KRAS* G12V | Leibovitz et al, 1976; Berg et al, 2017 |
| NCC-375 | 24, f, Korean | Adenocarcinoma, rectum, moderately differentiated | n/a | n/a | Kim et al, 2019 |
| SNU-1460 | 25, m, Korean | Adenocarcinoma, sigmoid colon, moderately differentiated | MSS | *KRAS* G13D<br>*TP53* R215P | Ku et al, 2010; Kim et al, 2019 |
| SNU-1826 | 15, m, Korean | Adenocarcinoma, cecal cancer, poorly differentiated | n/a | n/a | Kim et al, 2019 |
| SNU-2446 | 21, f, Korean | Adenocarcinoma, descending colon | n/a | n/a | Kim et al, 2019 |
| CCD 841 CoN | GW 21, f | Normal fetal colon | n/a | n/a | Thompson et al, 1985 |
| FHC | GW 13, m | Normal fetal colon | n/a | *TP53* R273H | Siddiqui and Chopra, 1984 |

*GW* gestational week.

(Rahman et al, 2021). Moreover, coating of the insert with extracellular matrix (ECM) is used to mimic the basement membrane and provides a more physiological and structural scaffold, and thus further supports cell polarization (Jain et al, 2022). Finally, the formation of a confluent monolayer on the membrane facilitates the formation of cell-cell junctions, essential for the establishment of apical–basal polarity which usually develops after certain days to weeks (Olejnik et al, 2003). Hence, transwell-based co-culture assays create an in vitro environment that accurately represents the polarized structure and function of colon epithelial cells. One major advantage is the creation of two compartments allowing to model an oxygen low apical ($< 1\%$ $O_2$) and an oxygen high ($> 10\%$ $O_2$) basolateral compartment, recreating $O_2$ concentrations also found under physiological conditions (Kim et al, 2022). This is warranted by either sealing the apical chamber from atmospheric oxygen, e.g., with a plug or liquid paraffin (Sasaki et al, 2020; Kim et al, 2022; Umehara and Aoyagi, 2023), or by placing the co-culture system in an anaerobic work station and using oxygenated medium for the basolateral chamber to supply the colon cells (Ulluwishewa et al, 2015). Moreover, the confluent cell monolayer helps to stop leakage of oxygen across the two compartments (Maier et al, 2018). The so-created apical-anaerobic, basolateral-aerobic atmosphere overcomes the problem of a limited lifespan of anaerobic bacteria often observed in other co-culture systems (Umehara and Aoyagi, 2023; Kim et al, 2022). However, due to the static nature of transwell-based models, co-cultivating time is mostly limited to <24 h, as they rapidly result in acidic medium due to bacterial fermentation products and bacterial overgrowth and are thus highly cytotoxic for the host cells (Penarete-Acosta et al, 2024;

Jalili-Firoozinezhad et al, 2019). Furthermore, the healthy intestinal mucosa is characterized by (1) an oxygen gradient from proximal to distal colon and from the submucosa to the gut lumen and (2) a mucous layer serving as efficient tissue barrier (Albenberg et al, 2014), which both cannot be accurately modeled in the transwell system. Altogether, transwell co-culture systems are a feasible approach to study short-term bacterial adhesion and invasion, while their limitation are the lack of the typical gut mucus layer as well as oxygen gradients, the dependence on artificial cell lines and rapid acidification due to missing medium exchange.

## Microfluidic co-culture models

Microfluidic devices add a new level of complexity to co-culture systems by providing constant flow. The application of flow has at least two major advantages: On the one hand, it allows longitudinal co-cultures due to constant medium exchange, preventing excessive bacterial growth and acidification. On the other hand, flow more accurately mimics the physiological conditions as it can reproduce the intestinal linear flow rate (~5–30 µm/s (Shin et al, 2019)) and thus exert shear stress on the colorectal epithelial cells which supports cellular differentiation and mucus production (Lindner et al, 2021). Mucus, in turn, affects bacterial adhesion and contact to virulence factors (Johansson et al, 2013).

Like transwell-based co-cultures, microfluidic devices typically create two separate channels, which can be adjusted to the different oxygen requirements of cells and bacteria. One example for such a set-up is the "anoxic-oxic interface-on-a-chip". Here, cells seeded on top of a semipermeable membrane are perfused with anoxic medium in the apical and oxygenated medium in the basolateral

channel, thus allowing co-cultures of colorectal cells and gut bacteria of up to a week (Shin et al, 2019). A variation of the typical two-chamber set-up is the microfluidic device invented by Mittal et al which consists of a middle chamber for colorectal cell culture connected via six linear channels with two side channels, through which medium or bacteria can be introduced (Mittal et al, 2023). This set-up creates a longitudinal concentration gradient in the middle chamber and allows studying the effects on cell growth via live-cell imaging.

Another more advanced system is the Host–Microbiota Interaction (HMI) module connected to a Simulator of the Human Intestinal Microbial Ecosystem (SHIME) gut simulator (Marzorati et al, 2014). SHIME consists of five specific fermenters, one for stomach, small intestine as well as ascending, transverse and descending colon, which are thought to recapitulate the physiological, chemical and microbiological properties of the individual gut compartments (Van De Wiele et al, 2015; Penarete-Acosta et al, 2024). Upon inoculation with fecal samples of healthy donors or EO-CRC patients, sampling, e.g., from the ascending colon of the SHIME can be used to obtain microbial communities representing this specific gut compartment for studying their interplay with colon epithelial cells in the connected HMI module (Van De Wiele et al, 2015). The module itself consists of a layer of CRC cells in the lower chamber, which is physically separated by a mucus-covered semipermeable membrane from the microbial communities injected into the upper chamber. On the one hand, this set-up provides the unique opportunity to study the interplay of colorectal cells with complex microbial communities and monitor the formation of a gut biofilm on the mucin layer. On the other hand, it prevents the direct interaction of host cells and the microbial communities and thus allows only the study of contact-independent tumor-microbiome crosstalk, e.g., by soluble factors or extracellular vesicles. An adaptation of the HMI module is the Human-Microbial Crosstalk (HuMiX) co-culture system which introduces an additional lower medium perfusion chamber below the colorectal cell layer (Shah et al, 2016), thus ensuring nutrient and oxygen supply of the cells from the basolateral side and modeling more accurately the structure of the human gut.

To simulate even more complex environments, several microfluidic approaches have been designed that expand tumor-microbiome co-cultures with additional cell populations. In the anaerobic "intestine-on-a-chip", for instance, CRC cells were seeded on top of a semipermeable membrane, which was coated on the underside with human intestinal microvascular endothelial cells (HIMECs). The addition of endothelial cells was shown to particularly enhance the barrier function, mucus production and villi development of the colon cells (Jalili-Firoozinezhad et al, 2019). Integrating immune cells into co-cultures permits to model even more encompassing cellular interactions. One example is the gut epithelium-microbe-immune (GuMI) system, which was developed for co-culturing primary human colon epithelial cells with the highly oxygen-sensitive gut commensal bacterium *Faecalibacterium prausnitzii*, antigen-presenting cells as well as CD4$^+$ T cells. Using this set-up, the authors analyzed the function of CD4$^+$ T cells in orchestrating the immune response triggered by commensal gut bacteria (Zhang et al, 2024). To better mimic the tumor microenvironment in EO-CRC, incorporating primary microenvironmental cells derived from young individuals could help to more accurately capture the cellular dynamics and host-microbiome interactions occurring in young patients.

In recent years, a magnitude of microfluidic devices has been developed with the above-mentioned representing only a selection for studying the interplay of microbiota and colorectal cells. While they extend co-culture times up to several days and enable the simulation of complex multicellular interactions, custom-made components are often not off-the-shelf items, lack standardization and require special fabrication, extended expertise and time for set-up.

## Colorectal organoids

Colorectal organoids, 3D "miniature" versions of the intestine, allow researchers to study direct interactions between gut microbes and the various cell types of the human intestine, i.e., enterocytes, goblet cells, stem cells, enteroendocrine cells, Tuft cells, M cells and Paneth cells. One approach is to generate organoids from either human induced pluripotent stem cells (iPSCs) or adult stem cells (ASCs) obtained from small tissue biopsies, peripheral blood cells or even a single hair pluck (Poletti et al, 2021; Staerk et al, 2010; Maherali et al, 2008). Another approach of increasing popularity are patient-derived organoids (PDOs) which can be generated from healthy or tumor tissue (Mebarki et al, 2018). As PDOs maintain both phenotypic and genetic characteristics of the original tissue (Engel et al, 2022), they can overcome the limitations of other in vitro models. In particular, the comparison of CRC cell lines and PDOs has revealed significant differences in drug responses, indicating that PDOs are more representative models for personalized anti-cancer drug screens than traditional 2D cell cultures (Usui et al, 2016; Vlachogiannis et al, 2018). Based on these encouraging observations, biobanks of tumor and normal colorectal organoids have been established for high-throughput drug screening (van de Wetering et al, 2015).

To study the interaction of intestinal organoids with specific microbiota, different approaches have been published. One is the co-culture of microbes with sheared organoids in 2D, another, more physiological, approach is the microinjection of specific bacterial strains or communities directly into the hypoxic lumen of a 3D organoid (Puschhof et al, 2021). While the latter allows the microbes to directly interact with the apical side of intestinal cells within the natural tissue architecture, oxygen and nutrient levels are hard to control inside the organoids and the lack of flow, and thus a fully mature mucous layer, are major limitations of the system compared to other more complex ex vivo models. Despite the lower oxygen concentration in the lumen of the organoids, in particular co-cultures with obligate anaerobes remain challenging. In 2018, Williamson et al presented a high-throughput system which injected cargo (mostly aerobic bacteria) into approx. 90 printed organoids per hour to circumvent the labor-intensive and variable manual microinjections (Williamson et al, 2018). Despite these advancements in automation and high throughput, the microinjection of bacteria remains technically difficult and time-consuming, necessitating specialized equipment and expertise. Another pitfall is that it destroys the integrity of the different organoid layers (Nikonorova et al, 2023). To overcome this, the apical-out method can be applied (Kakni et al, 2023). Here, the polarity of the organoids is inverted, resulting in facing their apical surface outwards, while maintaining the 3D architecture. Although this approach facilitates the direct interaction of gut microbiota with the apical surface (Co et al,

2021), it fails to replicate the low-oxygen microenvironment physiologically observed at the apical surface of the intestine. Thus, hypoxic or anaerobic culture conditions need to be applied when inoculating oxygen-sensitive bacteria. Moreover, due to their more complex nature, successful cultivation of organoids depends heavily on the right growth factors and conditioned media. Therefore, adapting cultivation conditions for obligate anaerobic bacteria and for different media requirements is challenging. Nevertheless, there are reports that co-cultivations are feasible for at least 72 h (Williamson et al, 2018). Although PDO-microbiome co-cultures have not been extensively described yet, first reports have adapted microfluidic systems for PDOs (Williamson et al, 2018; Jalili-Firoozinezhad et al, 2019). Integrating microbiome studies with PDO research will allow for a more comprehensive understanding of how gut microbiota may influence tumor behavior and treatment responses. By co-culturing PDOs with specific (young) microbial communities, researchers can investigate potential interactions that could inform dietary or probiotic interventions as adjunct therapies (Barbáchano et al, 2021).

In conclusion, intestinal organoids are extremely valuable for studying tumor-microbiome interactions in the natural gut tissue architecture with its heterogeneous cellular composition. The opportunity to generate PDOs from healthy young individuals or EO-CRC patients of different gender, ethnic or genetic background harbors the unique advantage of obtaining cells that allow personalized or matched cohort microbiome studies and mimic colon cells more closely than cell lines that have adapted to long-term 2D culture conditions. Although most organoid models lack microenvironmental components, there are first reports on the implementation of endothelial cells, fibroblasts or immune cells into organoid cultures (Holloway et al, 2020; Wallisch et al, 2023; Strating et al, 2023; Li et al, 2023) that could potentially address this constraint in the future. Further limitations are the lack of oxygen gradients, the dependence on microinjections for tumor-microbiome interaction studies and the challenging set-up. PDOs are particularly difficult to establish from certain tumor types (e.g., MSI, *BRAF*-mutated, or mucinous-like tumors) (Jalili-Firoozinez-had et al, 2019) and researchers frequently have to adapt the CRC standard organoid media composition (Sato et al, 2011).

### Other ex vivo approaches

Ex vivo models are based on human or animal tissue explants which are cultured for a limited timespan in vitro. Whole tissue segments maintain not only the physiological architecture and structural organization, but also inter- and intra-tissue heterogeneity and the diverse spatial relationships (Sivakumar et al, 2019). For studying barrier function and electrolyte transportation in living tissue, one possibility is to use an Ussing chamber, adapted to microfluidic setups. This approach consists of a small-, interconnected compartmentalized system that holds an intestinal tissue sample as a whole and allows for perfusion from the sample chamber to detection zones, measuring electrical properties using electrodes. It can be used for short-term experimental settings (generally few hours) measuring absorption and/or secretion (Ussing and Zerahn, 1951; Clarke, 2009). Alternatively, other tissue sources can be exposed within the living organism to the compound of interest (e.g., different microbiota) prior to sampling and introduction into the Ussing chamber (Shi et al, 2014; Chen et al, 2010).

Another short-term model, often used in pharmaceutical research, is the Everted Sac. Similar to the apical-out method used for organoids, a segment of intestinal tissue, mostly obtained from different animal models, is turned inside out (i.e., by everting it), thus exposing the mucosal surface to the external medium (Alam et al, 2012; Wilson and Wiseman, 1954). Advantages are the relatively large surface area, the presence of an epithelial mucus layer, high levels of paracellular transport as well as the possibility to study drug absorption in the different gut compartments (Alam et al, 2012). While so far not having been applied in EO-CRC research, the addition of selected gut microbiota to the culture medium could potentially create another ex vivo co-culture system to investigate gut microbiome interactions.

For mid- to long-term studies, organotypic slice cultures of dissected colon tissue can be employed and maintained for several days up to four weeks (Baydoun et al, 2017). Schwerdtfeger et al have already successfully employed this model to study gut segmental contractions in the presence of microbiota through the use of specific antibiotics (Schwerdtfeger et al, 2016). However, the primary disadvantage of this model is the aerobic conditions under which the experiments need to be conducted, leading to a selective survival of only facultative anaerobic bacteria. Recently, the development of so-called "mini-colons" has provided another promising ex vivo tool (Lorenzo-Martín et al, 2024). By introducing a light-sensitive Cre system into inducible $Apc^{fl/fl}Kras^{LSL-G12D/+}Trp53^{fl/fl}$ healthy colon organoids and seeding them in a hydrogel-patterned microfluidic device, a gut characteristic crypt- and lumen-like structure was replicated. Exposing the cells to blue light and doxycycline allowed a spatiotemporally controlled induction of tumorigenesis and its tracking in real-time at the single-cell level for several weeks without the need to disrupt the tissue for passaging (Lorenzo-Martín et al, 2024).

So far, tumor-microbiome interactions have mostly been studied at the level of single bacterial strains. However, the gut microbiome is complex and not only inhabited by pathogenic bacteria but also by commensals, impacting each other. To simulate complex microbial communities in the distinct gut segments, systems like the above-described SHIME or the Colon-on-a-Plate technology (Sánchez et al, 2023), simulating only the colorectal part, are available. Using these methods in combination with other in vitro or ex vivo host models, has the potential to evaluate the effects of diverse microbial communities on colorectal cells.

Although ex vivo models have not yet been extensively used in the context of EO-CRC, they provide a unique opportunity to study tumor-microbiome crosstalk in a controlled environment while maintaining much of the complexity and physiological relevance of in vivo systems. However, some of the approaches still suffer from a limited lifespan and all of them lack the systemic interactions present in a living organism. In particular, maintaining adequate nutrient and oxygen supply throughout the tissue without a functional blood vessel system can be technically challenging.

## Conclusion

To fully unravel and understand the role of the gut microbiome in the transformation of normal intestinal epithelial cells to invasive and metastatic CRC, in particular in young patients, the combination of different approaches (summarized in Fig. 2) is required as no currently available model fully recapitulates the complex interactions of this multifaceted system. In vivo experiments are a powerful tool for validating the biological significance of identified

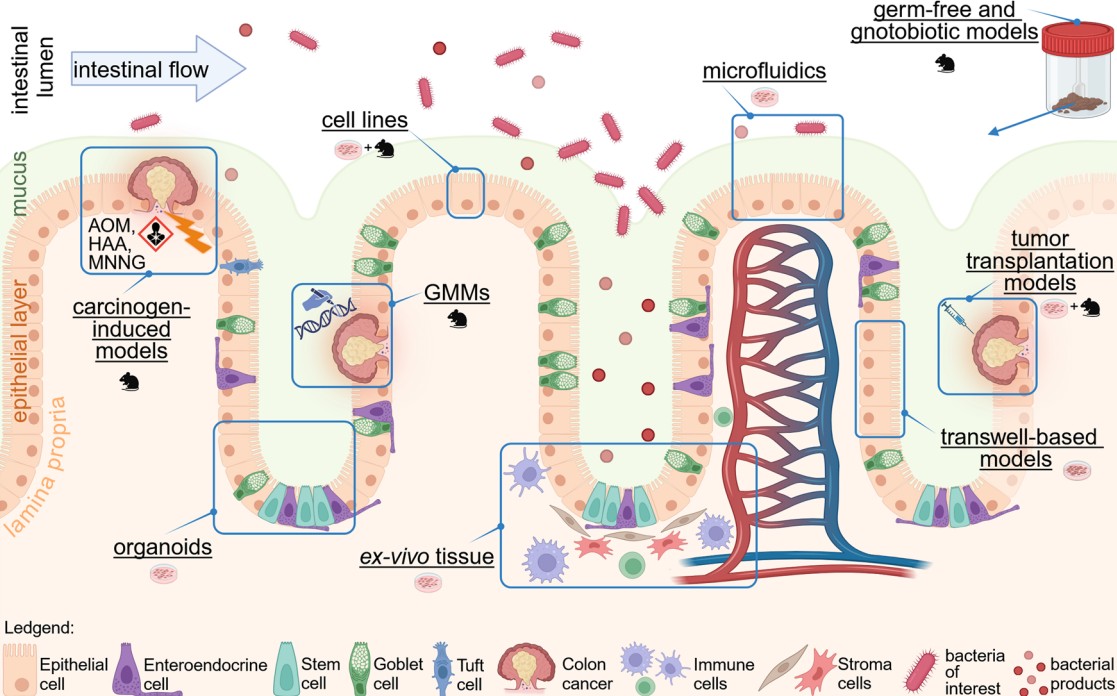

**Figure 2. Model systems to study tumor-microbiome interactions.**

EO-CRC development and growth upon contact with gut microbiota can be modeled via different in vivo, in vitro and ex vivo approaches. These include mouse models, cell line-based models as well as fecal microbiota transplantation and ex vivo tissues. All of them harbor unique characteristics and enable the study of specific aspects of tumor-microbiome interactions as exemplified in this figure. A detailed overview of the limitations and benefits of each model system is provided in Table 2.

targets and testing novel therapeutic approaches in a physiological environment. In particular, GMMs with a tamoxifen-inducible Cre-loxP system or the orthotopic injection of GMM-derived tumor organoids into immunocompetent young mice seem promising approaches to model EO-CRC in vivo. Additionally, several models exist to simulate specific features typically associated with EO-CRC such as FAP, MSI, or IBD. Combining these established models with the use of antibiotics and/or FMT offers the possibility to study the contribution of whole (young) microbiomes, or specific microbes, respectively, on EO-CRC development and growth in a living organism. However, due to the species barrier, lack of heterogeneity as well as a deficient adaptive immune system and missing tumor microenvironment in some models, caution should be taken when converting in vivo animal studies to humans.

Concomitantly, in vitro and ex vivo modeling of host-microbiome interactions presents valuable insights into the molecular mechanisms of the disease, despite not fully capturing the complexity of the human gut environment and microbiome. However, these models are crucial for understanding causal relationships, including the contribution of specific genes or microbiota associated with EO-CRC in controlled settings, often generating faster and more ethically acceptable results. Recently, advancements in organoid cultures and tissue explants have begun to emulate the biological complexity of the human gut and its microbiome, providing exciting novel opportunities for studying colon–microbiome interactions. Since they can be obtained specifically from EO-CRC patients, they harbor the unique possibility for personalized tumor-microbiome studies and might allow larger cohort studies to understand the higher incidence of EO-CRC in certain groups (e.g., males, African Americans).

Future developments should be focused on extending the co-cultures from single bacterial strains to whole microbial communities and microbiomes, including primary stroma and immune cells to decipher the specific contributions of young versus aged microenvironments and developing models that effectively capture the dynamic interactions between the microbiome and the metastatic niche. Taken together, a combination of the various approaches will improve our understanding of the mechanisms underlying EO-CRC development and hopefully result in the discovery of novel strategies for tumor prevention and therapy in young patients.

## Pending issues

1. Development of advanced models: Due to the limited translatability of the established 2D in vitro systems and available in vivo models, the development and refinement of ex vivo approaches (e.g., organoid or tissue slice cultures) are urgently needed to better recapitulate the biological complexity of the human gut and its microbiome. This particularly includes optimizing approaches that mimic the physiological environment of young patients.
2. Species barrier and human translation: Research must address the challenges posed by the species barrier in translating findings from animal models to human applications. Further studies are needed to identify strategies that enhance the relevance of animal models in understanding human CRC.
3. Longitudinal studies: Long-term studies are required to observe the dynamic changes in the gut microbiome and its relationship with CRC development over time, particularly in young

individuals. This could provide insights into the progression and potential prevention strategies for EO-CRC.

4. Molecular mechanisms of host-microbiome interactions: There is a critical need to elucidate the molecular mechanisms underlying host-microbiome interactions that contribute to the transformation of normal intestinal epithelial cells to invasive CRC. This includes identifying specific microbial species or metabolites that may play a role in tumorigenesis and could be targeted in microbiome editing approaches.

5. Metastasis: Models that capture the interaction between the metastatic niche and the microbiome are needed to better understand how the microbiome influences the metastatic process, including its role in shaping the microenvironment and driving mechanisms such as epithelial–mesenchymal transition (EMT) and tumor immune escape.

# Peer review information

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

## Acknowledgements

This project was supported by the German Federal Ministry of Education and Research (BMBF) (project 01KD2101F PerMiCCion) and the Open Access Publication Fund of the University of Muenster. KMR is a member of CiM-IMPRS, the joint graduate school of the Cells-in-Motion Interfaculty Center, University of Muenster, Germany, and the International Max Planck Research School-Molecular Biomedicine, Muenster, Germany. CK received funding from the Deutsche Forschungsgemeinschaft (DFG, German Research Foundation)—project no. 493624047 (Clinician Scientist CareerS Muenster). LCC is part of the Cancer-Specific Multidrug Nanocarriers for Colon Cancer Therapy and Imaging (CANACO) Consortium (German Cancer Aid, project no. 1843671) and acknowledges the Comprehensive Cancer Center Niedersachsen (CCC-N). All figures have been created in BioRender (Fig. 1: Richter, K. (2025) https://BioRender.com/v76d254; Fig. 2: Richter, K. (2025) https://BioRender.com/y37b747).

## Author contributions

**Katharina M Richter**: Writing—original draft; Writing—review and editing. **Marius Wrage**: Writing—original draft; Writing—review and editing. **Carolin Krekeler**: Writing—original draft; Writing—review and editing. **Tiago De Oliveira**: Writing—original draft; Writing—review and editing. **Lena-Christin Conradi**: Writing—review and editing. **Kerstin Menck**: Supervision; Writing—original draft; Writing—review and editing. **Annalen Bleckmann**: Supervision; Funding acquisition; Writing—review and editing.

## Funding

## Disclosure and competing interests statement

The authors declare no competing interests.

