## [Peer Review File · EMBO Molecular Medicine]

Model systems to study tumor-microbiome interactions in early-onset colorectal cancer

Katharina Richter, Marius Wrage, Carolin Krekeler, Tiago De Oliveira, Lena-Christin Conradi, Kerstin Menck, and Annalen Bleckmann

Corresponding author: Annalen Bleckmann (annalen.bleckmann@ukmuenster.de)

Review Timeline:

Submission Date:	30th Aug 24
Editorial Decision:	30th Sep 24
Revision Received:	19th Dec 24
Editorial Decision:	10th Jan 25
Revision Received:	13th Jan 25
Accepted:	24th Jan 25

Editor: Zeljko Durdevic

Transaction Report:

30th Sep 2024

Dear Dr. Bleckmann,

Thank you for the submission of your manuscript to EMBO Molecular Medicine. We have now received feedback from the three reviewers who agreed to evaluate your manuscript. As you will see from the reports below, the referees are positive about its interest and timeliness, however, they also raise serious criticisms that should be addressed in a revised manuscript. Further consideration of a revision that addresses reviewers' concerns in full will entail a second round of review. Particular focus should be given to:

- 1) Rewriting and restructuring the manuscript to better emphasize how existing understanding of EO-CRC biology and clinical features could be integrated into the development of the relevant models as suggested by the referee #2.
- 2) Implementing all the referee #1 and #3 suggestions.

I would also like to ask you to add the following items to your revised article:

- 1) Please upload figures as individual high-resolution files. Figure 2 appears too complicated, and the font is too small. Please consider dividing figure 2 into 2 figures that would represent in vitro and in vivo approaches, respectively. If BioRender was used to create the figures, please add following sentence to the figure legends: "Graphics were created with BioRender.com."
- 2) Please provide more detailed description of all figures in the legends.
- 3) Glossary: The glossary is meant to explain some of the terms used for laymen. Could you please identify terms that may need an "explanation"?
- 4) Pending issues: At the end of each article is a box highlighting issues that still need further studies and where research efforts should converge. Could you identify some pending issues?

I hope that the referees' comments do not prove too problematic to address and I look forward to reading your next version.

Yours sincerely,

Zeljko Durdevic

*** IMPORTANT INFORMATION ***

- 1) a .doc formatted version of the manuscript text (including Figure legends and tables)
- 2) Separate figure files
- 3) a letter INCLUDING the reviewer's reports and your detailed responses to their comments.

Also, and to save some time should your paper be accepted, please read below for additional information regarding some features of our research articles:

- 1) Glossary: EMBO Molecular Medicine articles will be accompanied by a glossary explaining some of the terms used for laymen. I identified the following:

_____, _____, _____

Could you please help us in identifying terms that may need an "explanation" other terms that we can add to the glossary.

2) For more information: This is a short list of related web links for further consultation by the readers. Could you identify some relevant ones? Examples are patient associations, OMIM related links, databases, authors websites, etc.

3) Pending issues: At the end of each article we will have a box highlighting issues that still need further studies and where research efforts should converge (we call this the Pending issues box). From my reading I would say:

but I can see there may be many more. Could you work on this as well?

4) Disclosure and competing interest statement: Please include a statement declaring any competing commercial interests in relation to your submitted work.

5) Please note that we now mandate that all corresponding authors list an ORCID digital identifier. This takes <90 seconds to complete. We encourage all authors to supply an ORCID identifier, which will be linked to their name for unambiguous name identification.

Currently, our records indicate that the ORCID for your account is 0000-0002-0863-9840.

Link Not Available

-

Thank you,

Zeljko Durdevic

***** Reviewer's comments *****

Referee #1 (Remarks for Author):

Introduction:

The introduction covers the rising incidence of EO-CRC and the potential role of the microbiome. However, it lacks depth in explaining why the microbiome might play a different role in EO-CRC compared to CRC in older populations. Given the focus of the review, I think this should be explained.

Suggestion: Provide a more detailed background on EO-CRC, focusing on how the microbiome's interaction with younger immune systems, lifestyles (e.g., diet, antibiotics use), or genetic predispositions might differ from older CRC cases. This would better justify the need for the review.

In Vivo Models:

This section provides an extensive overview of in vivo models, but seems like the critical assessment is somewhat limited. There is a lack of discussion about how well these models reflect the human condition of EO-CRC and the microbiome's role.

Suggestions:

- Clearly outline the strengths and weaknesses of each model concerning EO-CRC. Which models are best for short-term vs. long-term studies? Which can best replicate the human microbiome or immune interactions?
- Highlight more explicitly why GEMMs like ApcMin/ or other models might fall short for microbiome research, particularly in EO-

CRC. For example, emphasize the limitation that most tumors in these models develop in the small intestine rather than the colon.

· For carcinogen-induced models: maybe it's better to strengthen this point as the carcinogen-induced models are useful for understanding general CRC mechanisms, while they may not adequately reflect EO-CRC due to latency issues and lack of invasive behavior.

In vitro model and Ex vivo models:

This section is strong in its technical detail, but it seems like a little bit difficult to assess which models are most suitable for specific research questions. The discussion of transwell, microfluidic, and organoid models is well-done, but lacks a critical examination of their limitations, especially in terms of the microbiome's complexity.

Suggestions:

· The transwell models should include a more detailed discussion of their limitations when studying long-term microbiome interactions. For example, how do these models simulate anaerobic environments, and what are their limitations in representing the gut's oxygen gradients?

· The organoid section mentions their growing popularity, but there should be more emphasis on the technical challenges-such as maintaining gut microbiota in an organoid setup and the difficulty of microinjections. Additionally, discuss the potential for patient-derived organoids and their role in personalized microbiome studies for EO-CRC patients.

May be add a table that outlines important information of each models, such as advantages, disadvantage, applications, etc.

Specific comments:

add reference Spaander et al.: <https://www.nature.com/articles/s41572-023-00432-7>

Section on APC min mouse: link APCMin model to FAP en mention loss of heterozygosity while specifying kinetic of tumor development. Also mention that most tumors in APCMin mice develop in the small intestine, while human intestinal cancer mostly presents in the colon. Another limitation of the model is the fact that APCmin mice are not breeding effectively.

Section on cre based targeting: An alternative approach is to use floxed lines with an inducible VillinERT2Cre allele, and subsequent local injection of tamoxifen in specific regions of the colon to induce local tumor formation. Alternatively, tumor organoids derived from GEMMs can be locally injected in the colon of wild type congenic recipient strains.

Referee #2 (Remarks for Author):

This review attempts to address the timely topic of the increase of early-onset colorectal cancer (EOCRC), affecting people under 50 years old. The author touched about the lifestyle related risks as potential causes of EOCRC and mentioned its direct influences on the gut microbiome, which is also a rising field in colorectal cancer (CRC) carcinogenesis. The review is intended to give an overview about in-vivo, in-vitro and ex-vivo models that could be utilized to study two important aspects of CRC carcinogenesis.

Nevertheless, despite many emerging studies in the literature, this review fails to develop much needed description of how current CRC models could be tailored to be used to study the biological mechanism of EOCRC through its association with the gut microbiome communities. Referring to the title of the manuscript, the reader would expect to find how existing understanding of EOCRC biology and clinical features could be integrated into the development of the relevant models (in-vivo, in-vitro and ex-vivo). For instance, it is well describing that EOCRC is associated with microsatellite instability (MSI), lower BRAF mutation and higher association with consensus molecular subtype (CMS)-1. These characteristics could be mentioned, and relevant models could be discussed. For example, Villin CreER Mlh1fl/fl could be described as one of MSI in vivo model that should be explore further. EOCRC is also associated with metastasis, thus models that capture the interaction of metastatic niche with microbiome could be explored. In the manuscript, when discussing the in-vivo models, the authors seem to oversimplify the EOCRC carcinogenesis as the latency of the models. Perhaps it is worth to address if there are differences in latency of the same model in young and older mice.

For more general notes, despite the review provide in detail about the latest advancement of in-vitro CRC models to study gut microbiome, the description of the in-vivo models in this review is less comprehensive and not capturing the latest developments. This review will also benefit from expanding the literature included in the discussion.

Overall, the authors may consider revising this review by putting the effort to link the key aspect of EOCRC and gut microbiome to provide specific models to be used in further studies by the scientific communities. If not, the review just simply providing information about in-vivo, in-vitro and ex-vivo to study CRC-microbiome interaction in general which lower its originality.

Referee #3 (Remarks for Author):

In this review, the authors attempted to provide an overview of the model system to investigate tumor-microbiome interactions in early-onset colorectal cancer (EO-CRC). It presents a fairly comprehensive overview of the methods that could be applied to the study CRC, and also commented on their suitability for EO-CRC. The review could be further improved by addressing the following points:

1. Although the authors have commented the suitability of different mouse models for EO-CRC based on the duration of the model, one question arises is how could one extrapolate the age in mice to that of human? At what age is mice considered to be equivalent to ~40-50 year-olds? Many models discussed here, e.g. Apc mice, develop tumors in ~6 months. Given that mice can live up to 18-24 months (or more), I cannot see why these models are considered not suitable to study EO-CRC. Actually, one can argue that most mouse models used by researchers might be EO-CRC.
2. If one is aiming to decipher the role of microbiome-CRC tumor interplay, the authors should also provide an overview of FMT and germ-free models that play vital roles in elucidating the role of microbiome in disease pathogenesis. In addition, the authors have put great lengths to discuss the biological age from the viewpoint of host or tumor cells? How about the microbiome? Any studies showing that young or old microbiomes have differential impact on tumorigenesis?
3. The authors need to cite more studies that have compared the effect of engrafting tumor cells to young mouse vs aged mouse hosts. Beyond the cell themselves, the host age is thought to impact tumorigenesis.
4. Aging also brought about potential changes in the tumor immune microenvironment, which is pivotal to tumor growth and response to therapy. Please summarize the literature in this area.
5. Many of the in vitro models described is not that relevant to EO-CRC, perhaps with the exception to the use of cell lines/organoids from EO-CRC patients. Can the authors elaborate how could one better mimic the aged tumor microenvironment?
6. A brief overview of the characteristics (genetics/epigenetics/microbiome) of EO-CRC compared to CRC in aged subjects is recommended to give the readers an idea of the particularities EO-CRC.

Referee #1 (Remarks for Author):**Introduction:**

The introduction covers the rising incidence of EO-CRC and the potential role of the microbiome. However, it lacks depth in explaining why the microbiome might play a different role in EO-CRC compared to CRC in older populations. Given the focus of the review, I think this should be explained. Suggestion: Provide a more detailed background on EO-CRC, focusing on how the microbiome's interaction with younger immune systems, lifestyles (e.g., diet, antibiotics use), or genetic predispositions might differ from older CRC cases. This would better justify the need for the review.

Thank you for this important remark. We thoroughly revised the introduction and added more background information about EO-CRC, highlighting its differences (e.g. genetics, lifestyle) from LO-CRC (summarized in new Table 1). Moreover, we have added a paragraph about the microbiome changes in young patients and that aging affects both the microbiome and the immune system, which are in constant exchange. Since a detailed discussion of this topic would exceed the focus of this review, we have directed the interested readers to a comprehensive review on this topic.

In Vivo Models:

This section provides an extensive overview of in vivo models, but seems like the critical assessment is somewhat limited. There is a lack of discussion about how well these models reflect the human condition of EO-CRC and the microbiome's role. Suggestions:
-Clearly outline the strengths and weaknesses of each model concerning EO-CRC. Which models are best for short-term vs. long-term studies? Which can best replicate the human microbiome or immune interactions?

We thank the reviewer for this comment. As suggested, we have prepared new Table 2, which summarizes the strengths and weaknesses of each model system and gives information about the potential experimental timeframe. Moreover, we have critically revised the *in vivo* chapter to highlight which of the models are, in our opinion, suitable for EO-CRC research and we have added some recent advances that allow to study immune interactions in the respective settings.

-Highlight more explicitly why GEMMs like ApcMin/ or other models might fall short for microbiome research, particularly in EO-CRC. For example, emphasize the limitation that most tumors in these models develop in the small intestine rather than the colon.

We agree with the reviewer that this is an important limitation and have added it to the revised GMM section.

-For carcinogen-induced models: maybe it's better to strengthen this point as the carcinogen-induced models are useful for understanding general CRC mechanisms, while they may not adequately reflect EO-CRC due to latency issues and lack of invasive behavior.

We agree with the reviewer that the sole use of carcinogen-induced models might not be the best option for EO-CRC. We have revised the text as follows:

“Overall, carcinogen-induced CRC models offer several advantages that make them valuable tools for understanding general CRC mechanisms; however, they do not adequately reflect EO-CRC due to the high latency in tumor development, the lack of invasive behaviour and the effect of the carcinogens on the gut microbiome.”

In vitro model and Ex vivo models:

This section is strong in its technical detail, but it seems like a little bit difficult to assess which models are most suitable for specific research questions. The discussion of transwell, microfluidic, and organoid models is well-done, but lacks a critical examination of their limitations, especially in terms of the microbiome's complexity. Suggestions:

-The transwell models should include a more detailed discussion of their limitations when studying long-term microbiome interactions. For example, how do these models simulate anaerobic environments, and what are their limitations in representing the gut's oxygen gradients?

We thank the reviewer for this valuable feedback. We have revised the manuscript accordingly and included more details about the oxygen conditions and/or lack of oxygen gradients in transwell systems and organoids.

-The organoid section mentions their growing popularity, but there should be more emphasis on the technical challenges-such as maintaining gut microbiota in an organoid setup and the difficulty of microinjections. Additionally, discuss the potential for patient-derived organoids and their role in personalized microbiome studies for EO-CRC patients.

As requested, we have carefully revised the organoid section and included a more detailed description of the current challenges and the potential of organoids for personalized drug response and microbiome studies in EO-CRC patients.

-May be add a table that outlines important information of each models, such as advantages, disadvantage, applications, etc.

We agree with the reviewer that such a table would be highly valuable for the reader and have included it as new Table 2.

Specific comments:

Add reference Spaander et al.: <https://www.nature.com/articles/s41572-023-00432-7>

We have included the reference in the revised manuscript.

Section on APC min mouse: link APCMin model to FAP and mention loss of heterozygosity while specifying kinetic of tumor development. Also mention that most tumors in APCMin mice develop in the small intestine, while human intestinal cancer mostly presents in the colon. Another limitation of the model is the fact that APCmin mice are not breeding effectively.

We thank the reviewer for these important points. We have thoroughly revised the GMM chapter to highlight the link of the APC^{Min} model to FAP, include information regarding the timing of adenoma formation as well as the limitations of the model regarding tumor location and animal breeding.

Section on cre based targeting: An alternative approach is to use floxed lines with an inducible VillinERT2Cre allele, and subsequent local injection of tamoxifen in specific regions of the colon to induce local tumor formation. Alternatively, tumor organoids derived from GEMMs can be locally injected in the colon of wild type congenic recipient strains.

Following the suggestion of the reviewer, we have included a paragraph about the Cre-loxP system, including the VillinERT2Cre approach, in the GMM chapter as well as a paragraph about the injection of GMM-derived tumors into young mice in the tumor transplantation chapter.

Referee #2 (Remarks for Author):

This review attempts to address the timely topic of the increase of early-onset colorectal cancer (EOCRC), affecting people under 50 years old. The author touched about the lifestyle related risks as potential causes of EOCRC and mentioned its direct influences on the gut microbiome, which is also a rising field in colorectal cancer (CRC) carcinogenesis. The review is intended to give an overview about in-vivo, in-vitro and ex-vivo models that could be utilized to study two important aspects of CRC carcinogenesis.

Nevertheless, despite many emerging studies in the literature, this review fails to develop much needed description of how current CRC models could be tailored to be used to study the biological mechanism of EOCRC through its association with the gut microbiome communities. Referring to the title of the manuscript, the reader would expect to find how existing understanding of EOCRC biology and clinical features could be integrated into the development of the relevant models (in-vivo, in-vitro and ex-vivo). For instance, it is well describing that EOCRC is associated with microsatellite instability (MSI), lower BRAF mutation and higher association with consensus molecular subtype (CMS)-1. These characteristics could be mentioned, and relevant models could be discussed.

We thank the reviewer for the critical evaluation of our work. In the revised introduction, we now provide a more in-depth description of the key features of EO-CRC, distinguishing it from LO-CRC, including the higher frequency of MSI and CMS-1 as well as lower frequency of BRAF mutations. Additionally, we have summarized the differences in new Table 1. We have also critically revised the main text to highlight the applicability of each model system for EO-CRC research.

-For example, Villin CreER Mlh1fl/fl could be described as one of MSI in vivo model that should be explore further.

We thank the reviewer for pointing our attention to this model. We have added it in the GMM chapter: "To study the relationship between MSI and microbiome changes in EO-CRC, the use of VillinCreER Mlh1fl/fl mice (Leach et al, 2021) could be a promising model, although it may be necessary to combine the strain with other tumor mutation-carrying GMMs or with a carcinogen-induced model to reduce the latency for tumor development."

-EOCRC is also associated with metastasis, thus models that capture the interaction of metastatic niche with microbiome could be explored.

We thank the reviewer for this suggestion. We have critically revised the main text and added known models for studying mechanisms of EO-CRC invasion and metastasis in all suitable sub-chapters. As currently the knowledge about the microbiome in the metastatic niche is limited, we have added the topic to the "pending issues" paragraph, as suggested by the editorial team.

-In the manuscript, when discussing the in-vivo models, the authors seem to oversimplify the EO-CRC carcinogenesis as the latency of the models. Perhaps it is worth to address if there are differences in latency of the same model in young and older mice.

We thank the reviewer for this suggestion. We have included additional studies on tumor growth in young vs. old animals, which have been mainly performed using tumor or fecal microbiota transplantation approaches or the tamoxifen-inducible Cre-LoxP system in GMMs. In contrast, the most widely-used carcinogen AOM is typically administered in a total of six doses to young animals (6-8 weeks) and colon tumors are usually seen after 24 weeks in old animals (30 weeks) (PMID: 35885015). Therefore, we do think that the usefulness of this model for EO-CRC studies is limited by its long latency, next to its predisposition for colitis-induced tumor development.

-For more general notes, despite the review provide in detail about the latest advancement of in-vitro CRC models to study gut microbiome, the description of the in-vivo models in this review is less comprehensive and not capturing the latest developments. This review will also benefit from expanding the literature included in the discussion.

We thank the reviewer for drawing our attention to this important point. We have carefully reviewed the latest literature on *in vivo* models and added a new subchapter on germ-free and gnotobiotic models, including recent advances in microbiome editing, which had been missing so far. Additionally, we have expanded our discussion by including studies with particular relevance for EO-CRC.

- Overall, the authors may consider revising this review by putting the effort to link the key aspect of EO-CRC and gut microbiome to provide specific models to be used in further studies by the scientific communities. If not, the review just simply providing information about in-vivo, in-vitro and ex-vivo to study CRC-microbiome interaction in general which lower its originality.

We agree with the reviewer that this is an excellent idea. We have critically revised our manuscript to highlight approaches that are suitable for studying specific aspects of colon-microbiome interactions in the context of EO-CRC (e.g. germ-free models, Villin-CreER Mlh1fl/fl mice, DSS treatment). To this end, we have included new Table 1 to summarize the key features of EO-CRC and new Table 2 to summarize the main benefits and limitations of the individual models in the context of EO-CRC to help the reader in choosing the most appropriate model for their specific question.

Referee #3 (Remarks for Author):

In this review, the authors attempted to provide an overview of the model system to investigate tumor-microbiome interactions in early-onset colorectal cancer (EO-CRC). It presents a fairly comprehensive overview of the methods that could be applied to the study CRC, and also commented on their suitability for EO-CRC. The review could be further improved by addressing the following points:

1. Although the authors have commented the suitability of different mouse models for EO-CRC based on the duration of the model, one question arises is how could one extrapolate the age in mice to that of human? At what age is mice considered to be equivalent to ~40-50 year-olds? Many models discussed here, e.g. Apc mice, develop tumors in ~6 months. Given that mice can live up to 18-24 months (or more), I cannot see why these models are considered not suitable to study EO-CRC. Actually, one can argue that most mouse models used by researchers might be EO-CRC.

We agree with the reviewer that this is an important information and have added the information on age translation to the introductory paragraph of our *in vivo* part. Although middle aged mice (10-14 months old) could theoretically be considered for experiments on EO-CRC, it is known that its cellular maturation is happening approximately 150x faster. Therefore, to avoid the influence of additional co-variables on the obtained results (such as, hormone release, intestinal epithelial cells senescence, spontaneous mutations, etc.), the use of younger mice (up to 6 months age) is advised for EO-CRC research. This is especially important for experiments involving colorectal carcinogenesis, where the intestinal cellular turnover is extremely fast (PMID: 31995731). Additionally, it has been already shown that older mice show bigger microbial shifts, again adding extra covariates to analysis (PMID: 25520805; PMID: 16204576, PMID: 24467949). This is why we have concentrated on studies that utilized mice up to 6 months of age as young mice.

2. If one is aiming to decipher the role of microbiome-CRC tumor interplay, the authors should also provide an overview of FMT and germ-free models that play vital roles in elucidating the role of microbiome in disease pathogenesis. In addition, the authors have put great lengths to discuss the biological age from the viewpoint of host or tumor cells? How about the microbiome? Any studies showing that young or old microbiomes have differential impact on tumorigenesis?

We thank the reviewer for pointing our attention to this missing information and the opportunity to revise the text accordingly. We have added a new sub-chapter on germ-free and gnotobiotic models, including FMT studies. Here, we also included some studies that indeed revealed functional differences in the gut microbiome of young and old mice: "FMT from old (18 months), but not young (11-14 weeks), conventional mice into young (12-14 weeks) germ-free mice results in a low grade inflammation (PMID: 29163474) and can thus predisposition the animals for carcinogenesis. Indeed, Crossland *et al.* performed FMT from young (6 weeks) and old (72 weeks) mice into young (8 weeks) recipient mice. After initiation of tumorigenesis with AOM, the authors showed that FMT from older mice led to higher colonic cell proliferation, inflammation and tumor abundance, indicating age-dependent differences in the gut microbiome and its impact on tumorigenesis (PMID: 38031252)." Furthermore, we

have expanded our introduction and added more information about the effect of aging on the host as well as the microbiome.

3. The authors need to cite more studies that have compared the effect of engrafting tumor cells to young mouse vs aged mouse hosts. Beyond the cell themselves, the host age is thought to impact tumorigenesis.

We thank the reviewer for this remark and agree that the age of the host is an important aspect when considering engraftment efficiency, tumor formation and growth. We have revised the tumor transplantation chapter to accommodate these points as follows:

„It has been shown that the injection of MC38 cells into young C57BL/6 mice (3-4 months) resulted in a faster and more aggressive tumor growth compared to middle-aged animals (12-14 months). The different growth rates were not observed in immune-deficient young versus old RAG1null mice, underlining the central role of the immune system for CRC tumorigenesis (PMID: 6582296).“

„The preferred age of host mice for engraftment is 5-6 weeks as these mice provide a stable and conducive environment for tumor formation and growth (PMID: 19052619). Additionally, also in our experience with CRC xenografts, implantation in young NRG animals shows better engraftment than in old ones. We hypothesize that this is again due to better microenvironment implantation conditions, such as good perfusion and faster cellular turnover during the regeneration process after implantation (PMID: 39007350).“

4. Aging also brought about potential changes in the tumor immune microenvironment, which is pivotal to tumor growth and response to therapy. Please summarize the literature in this area.

We agree with the reviewer that this is an important point and have added a new paragraph in the introduction which reads as follows:

“EO-CRC cases show a stronger microbial-host interaction at the tumor site compared to LO-CRC (PMID: 37967575). This is not surprising, as, on the one hand, the function of the immune system deteriorates with age, a process called immunosenescence (PMID: 33658390). This involves the exhaustion of progenitor cells, reduced production of adaptive immune cells and altered secretion of matrix metalloproteinases and pro-inflammatory cytokines (e.g. IL1 β and TNF α), ultimately resulting in a low-level inflammation and a dysfunctional immune response to pathogens (PMID: 22972935; PMID: 18222733; PMID: 15004143). On the other hand, aging not only affects the host but is also observed in the gut microbiome itself where it causes age-related dysbiosis with implications for the host immune system (comprehensively reviewed in PMID: 33875817). The comparison of the immune landscapes in EO-CRC and LO-CRC is a matter of current research (PMID: 36921601,

PMID: 38152402, PMID: 36900249) and will hopefully improve our understanding of the mutual effects of host-microbiome interactions during aging and their functional relevance for EO-CRC development.”

5. Many of the *in vitro* models described is not that relevant to EO-CRC, perhaps with the exception to the use of cell lines/organoids from EO-CRC patients. Can the authors elaborate how could one better mimic the aged tumor microenvironment?

We agree with the reviewer that it is important to not only focus on the tumor cell themselves, but also consider the effect of aging on the microenvironment, in particular the immune system. However, as this manuscript is focused on EO-CRC, we think that it is more suitable to discuss how analyses could be performed in young instead of aged microenvironments. In response to the reviewer's suggestion, we have added the organoid model of Felchle *et al.* which will enable the analysis of the direct interactions between the tumor microenvironment and the microbiome in immunocompetent young mice *in vivo* (doi: 10.1016/j.ijrobp.2023.10.008) and we have highlighted the importance of selectively incorporating primary microenvironmental cells derived from young individuals for *in vitro* co-cultures or use the available *ex vivo* models which include primary microenvironments (e.g. whole tissue explants from EO-CRC patients).

6. A brief overview of the characteristics (genetics/epigenetics/microbiome) of EO-CRC compared to CRC in aged subjects is recommended to give the readers an idea of the particularities EO-CRC.

We agree with the reviewer. As explained above, we have added new Table 1, which summarizes the key features of EO-CRC compared to LO-CRC.

As you will notice, we have revised the manuscript based on the suggestions and addressed all comments made by the reviewer. We appreciate your thorough evaluation of our work and thank you for giving us the opportunity to resubmit the revised manuscript, which, as we strongly hope, will now be acceptable for publication.

Sincerely yours,

Prof. Dr. med. Annalen Bleckmann
Professorin für Internistische Onkologie
Direktorin WTZ Netzwerkpartner Münster
Bereichsleitung Internistische Onkologie Medizinische Klinik A

10th Jan 2025

Dear Dr. Bleckmann,

Thank you for the submission of your manuscript to EMBO Molecular Medicine. I am pleased to inform you that we will be able to accept your manuscript pending the following final amendments:

- 1) Authors: E-mail correspondence to Kerstin Menck could not be delivered. Please update their e-mail addresses and make sure to enter correct e-mail addresses for all authors in our submission system.
- 1) Figures: Please remove figures from the main manuscript file and place their legends after References.
- 2) Tables: Please move all tables to the end of the manuscript, after figure legends.
- 3) Funding: Please make sure all funding sources are entered in both the Acknowledgement and in our submission system.
- 4) Please remove "For more information".
- 5) As part of the EMBO Publications transparent editorial process initiative EMBO Molecular Medicine will publish online a Review Process File (RPF) to accompany accepted manuscripts. This file will be published in conjunction with your paper and will include the anonymous referee reports, your point-by-point response and all pertinent correspondence relating to the manuscript. Let us know whether you agree with the publication of the RPF.

You can submit your revised files by logging onto our online manuscript tracking system or simply follow this link:

<https://embomolmed.msubmit.net/cgi-bin/main.plex>

I look forward to receiving the revised version of your manuscript.

Yours sincerely,

Zeljko Durdevic

*** IMPORTANT INFORMATION ***

- 1) a .doc formatted version of the manuscript text (including Figure legends and tables)
- 2) Separate figure files
- 3) a letter INCLUDING the reviewer's reports and your detailed responses to their comments.

Also, and to save some time should your paper be accepted, please read below for additional information regarding some features of our research articles:

- 1) Glossary: EMBO Molecular Medicine articles will be accompanied by a glossary explaining some of the terms used for laymen. I identified the following:

_____, _____, _____

Could you please help us in identifying terms that may need an "explanation" other terms that we can add to the glossary.

- 2) Pending issues: At the end of each article we will have a box highlighting issues that still need further studies and where research efforts should converge (we call this the Pending issues box). From my reading I would say:

but I can see there may be many more. Could you work on this as well?

3) Disclosure and competing interest statement: Please include a statement declaring any competing commercial interests in relation to your submitted work.

4) Please note that we now mandate that all corresponding authors list an ORCID digital identifier. This takes <90 seconds to complete. We encourage all authors to supply an ORCID identifier, which will be linked to their name for unambiguous name identification.

Currently, our records indicate that the ORCID for your account is 0000-0002-0863-9840.

Link Not Available

-

Thank you,

Zeljko Durdevic

***** Reviewer's comments *****

Referee #1 (Remarks for Author):

I am happy with the changes and think it is acceptable for publication now

Referee #3 (Remarks for Author):

Is suitable for publication

The authors addressed the remaining editorial issues.

24th Jan 2025

Dear Dr. Bleckmann,

We are pleased to inform you that your manuscript is accepted for publication and is now being sent to our publisher to be included in the next available issue of EMBO Molecular Medicine.

Your manuscript will be processed for publication by EMBO Press. It will be copy edited and you will receive page proofs prior to publication. You will soon be contacted by Springer Nature to sign your publishing license. When you login to the customer service website, please use the following token to waive the article publication charges. Should you experience any difficulty, please email publishing@embo.org.
